# Rectifying Group Irregularities in Explanations for Distribution Shift

## Abstract

It is well-known that real-world changes constituting distribution shift adversely affect model performance. How to characterize those changes in an interpretable manner is poorly understood. Existing techniques take the form of shift explanations that elucidate how samples map from the original distribution toward the shifted one by reducing the disparity between the two distributions. However, these methods can introduce group irregularities, leading to explanations that are less feasible and robust. To address these issues, we propose Group-aware Shift Explanations (GSE), an explanation method that leverages worst-group optimization to rectify group irregularities. We demonstrate that GSE not only maintains group structures, but can improve feasibility and robustness over a variety of domains by up to 20% and 25% respectively.

## 1 Introduction

Classic machine learning theory assumes that training and testing data are sampled from the same distribution (Kearns & Vazirani, 1994). Unfortunately, distribution shifts infringe on this requirement and can drastically change a model's behavior (Kurakin et al., 2018). For instance, training a model on data collected from one hospital may result in inaccurate diagnoses for patients from other hospitals due to variations in medical equipment (Zech et al., 2018). Similarly, shifts from day to night or from clear to rainy weather are obstacles for autonomous driving (Wang et al., 2020).

When such a distribution shift occurs, it is often useful to understand *why* and *how* the data changed, independent of the model (Rabanser et al., 2019). For example, suppose a doctor observes their medical AI model's performance degrading. Before modifying the model, the doctor should first understand how their patient data changed (Subbaswamy & Saria, 2020). Similarly, a self-driving engineer would have an easier time adapting their system to a new environment if it was known that the shift resulted from changing weather conditions (manot AI, 2022).

The predominant method to understand a distribution shift is a *shift explanation* (Kulinski & Inouye, 2023). A shift explanation maps the original distribution (the source) to the shifted one (the target) to reduce their disparity. For example, Kulinski & Inouye (2023) find a direct mapping of points from the source toward the target via optimal transport (Peyré et al., 2017) and its variant, $K$-cluster transport. Another approach is to use counterfactual explanation methods such as DiCE (Mothilal et al., 2020) to explain a classifier between the source and target distributions.

State-of-the-art shift explanations seek to optimize global objectives, e.g. minimizing the difference between the target and the mapped source distribution (Kulinski & Inouye, 2023). However, these optimal mappings are not necessarily good explanations: they may be infeasible or lack robustness to source perturbations.

As an example, Figure 1 shows explanations from existing work that we learned to map individuals with low income (source distribution) to individuals with high income (target distribution) in the Adult dataset (Blake, 1998). Such explanations can help reveal insights about income inequalities that enable a policymaker to propose better policies or an individual to understand how to increase their income. At a dataset level, we see in Figure 1a that $K$-cluster transport can produce a shift explanation that effectively maps the source distribution to the target, resulting in an 86.3% reduction in the Wasserstein distance between these two distributions. However, upon closer inspection, this explanation shifts a majority Black cluster to a majority White cluster. Focusing on the Black racial

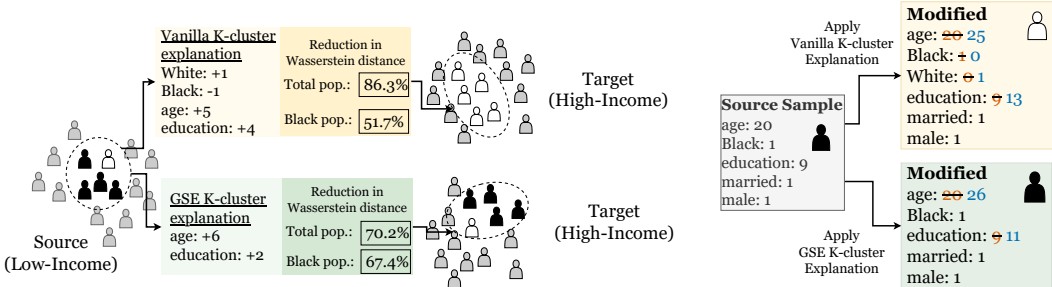

(a) Cluster level shift explanation.  (b) Instance level shift explanation.

Figure 1: (a) Compares shift explanations we learned from low to high-income populations using Vanilla and GSE $K$-cluster transport methods on the Adult dataset. Vanilla alters the race of a predominantly Black subpopulation, while GSE maintains Black and White subpopulations by slightly adjusting age (+6 years) and maximum education level (+2 years). (b) Illustrates instance-level explanations we learned; Vanilla changes race while our method does not. Full results in Appendix F.1.

subpopulation of the source and target, the explanation only decreases the Wasserstein distance by 51.7%. Such an explanation is not useful if such changes to race are considered infeasible[1].

Our key insight to achieving high-quality shift explanations is to steer the generated explanations to respect subpopulations, or **groups**, in the data. Since groups are highly context-specific, we seek an approach that is general and still produces overall good explanations. In our running example, assuming race-based grouping, such an approach should yield a mapping that minimizes disruptions to the groups while maximizing overall fitness. As depicted at the bottom of Figure 1a, we achieve such a mapping using the *same* underlying K-cluster transport method, that increases the reduction of Wasserstein distance between source and target samples from 51.7% to 67.4% within the Black subpopulation, and has a small impact on the reduction of Wasserstein distance between the overall source and target populations (from 86.3% to 70.2%).

To this end, we propose Group-aware Shift Explanations (GSE), an explanation method for distribution shift that preserves groups, equivalently conditional densities, in the data. We develop a unifying framework that allows us to apply GSE to heterogeneous methods for producing shift explanations including both optimal transport and counterfactual explanation methods, making them maintain group structures and enhancing their feasibility and robustness. Through extensive experiments over a wide range of tabular, language, and image datasets, we demonstrate that GSE not only maps source samples closer to target samples belonging to the same group, thus preserving group structure, but also boosts the feasibility and robustness by up to 23% and 42% respectively.

Our main contributions are summarized as follows:

1. We identify group irregularities as a class of problems that can adversely affect the quality of shift explanations, and we validate their existence empirically and theoretically.
2. We propose Group-aware Shift Explanations (GSE) to rectify group irregularities when explaining distribution shift, and enhance the feasibility and robustness of the shift explanations simultaneously, which are justified theoretically.
3. We propose a general framework to unify heterogeneous shift explanation methods, allowing the use of GSE for a wide range of shift explanation methods and domains.
4. We empirically demonstrate over a diverse set of datasets how GSE maintains group structures and enables more feasible and robust shift explanations.

## 2 MOTIVATION

Distribution shift is any change from an initial (source) to a different (target) distribution. We follow Kulinski & Inouye (2023) to define a shift explanation as a mapping from the source to the target distribution. For instance, Figure 1 shows a $K$-cluster explanation from our experiments which maps the source to the target distribution by changing race among other changes. In this section, we empirically identify issues with all existing shift explanations in terms of group irregularities.

---

[1]We note that the race attribute is not generally infeasible and decisions of infeasibility are left to the user.

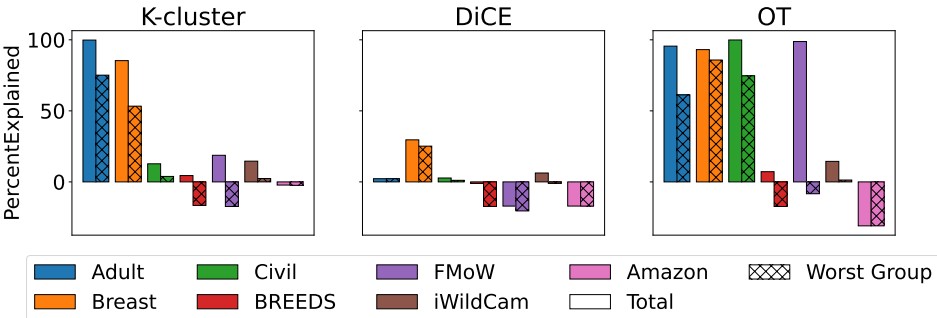

Figure 2: Overall vs. subgroup Wasserstein distance reduction from shift explanations. Hatched bars depict the minimum Wasserstein distance reduction within subgroups, contrasting with the solid bars representing overall reduction. PercentExplained signifies the percentage reduction in Wasserstein distance between source and target data. Across datasets and distribution shift explanation methods, the worst subgroup's Wasserstein distance reduction (hatched) is often markedly less than the overall reduction (solid).

## 2.1 GROUP IRREGULARITIES IN EXISTING SHIFT EXPLANATIONS

To find a shift explanation, state-of-the-art methods primarily minimize the disparity between the source and the target distribution. For example, $K$-cluster transport minimizes an objective depending on the Wasserstein distance between the source and the target distribution. However, this is not sufficient for finding high-quality explanations. Figure 1 shows such an example with $K$-cluster explanations where a majority Black subpopulation of the source gets mapped to a majority White subpopulation of the target. In this case, the overall Wasserstein distance is reduced by 86.3%, but the Wasserstein distance for the Black subpopulation is reduced much less in Figure 1a. Figure 2 shows that this problem is pervasive across the datasets and shift explanations we considered.

**Impact on Explanation Feasibility**   Shift explanations which break apart groups of the data can also be overall *infeasible*. Feasibility is a measure of how useful an explanation is to a downstream user, quantified by the percent of the source samples it is useful for. For instance, in Figure 1, the race attribute may be less actionable than others, so a $K$-cluster explanation which modifies the race attribute would be useless for a policymaker who designs policies to help increase the income of the low-income population. Overall, the $K$-cluster explanation from our experiments shown in Figure 1 is only feasible for 21.0% of the source distribution, meaning that 79.0% of the source samples have their race changed by the shift explanation. Later, we show how our method, which rectifies these group irregularities, results in more feasible explanations for the overall source distributions.

**Impact on Explanation Robustness**   Group irregularities can also reduce *robustness*, meaning that small changes to a source distribution result in large changes to the shift explanation. Figure 3 shows an example of poor explanation robustness from our experiments on the Adult and Civil Comments datasets. In Figure 3a, a small perturbation to the source distribution leads to the explanation modifying the race feature for a cluster of the data. Figure 3b shows a shift explanation that maps a non-toxic sample relating to medicine into the target distribution of toxic sentences. After a small perturbation, the explanation maps the same sample by adding the words "shooter" and "stupid" which is an unfeasible change since it changes the topic of the sample to violence. Ideally, we want a shift explanation to be robust to very small changes to the source distribution since it should explain general behavior instead of relying on minute details of a distribution.

Lastly, to formally validate that group irregularities are a fundamental problem, we study a simple 1D setting in Section 3.5 where we show that group irregularities always exist for explanations that optimize the reduction in Wasserstein distance between the source and target.

## 3   GROUP-AWARE SHIFT EXPLANATIONS (GSE)

In this section, we discuss our method and formalize the notions of feasibility and robustness introduced in Section 2 as metrics for evaluating shift explanations.

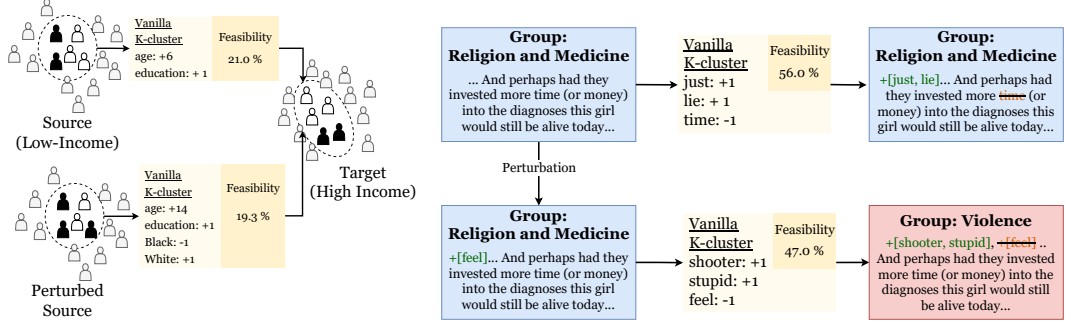

(a) Adult dataset.  (b) Civil Comments dataset.

Figure 3: Poor explanation robustness illustrated. Even if an explanation is feasible for a subpopulation (top), small perturbations to the source distribution can make it become infeasible (bottom). (b) Illustration of explanation robustness issues in the Civil Comments dataset, shifting from non-toxic to toxic text using KMeans-defined groups. More details on the group derivation are in Appendix E.

## 3.1 PRELIMINARIES ON $K$-CLUSTER TRANSPORT AND PERCENTEXPLAINED (PE)

The shift explanations produced by $K$-cluster transport can be denoted by a mapping function $M(x; \theta_x)$. The function $M(x; \theta_x)$ maps a source sample $x$ towards the target distribution by a distance of $\theta_x$, which is a learnable parameter. As the name $K$-cluster transport suggests, all the source samples are grouped into a set of clusters, $C$, with the $K$-means clustering algorithm, and within one cluster, $c \in C$, all the samples share the same $\theta_c$. Therefore, the mapping function for $K$-cluster transport is formulated as follows:

$$M(x; \theta) = x + \sum_{c \in C} \mathbf{1}_{x \in c} \theta_c, \text{ in which, } \theta = \{\theta_c | c \in C\}.$$

**Optimizing $\theta$ for $K$-cluster transport.** According to Kulinski & Inouye (2023), $\theta$ is solved by maximizing PercentExplained (PE). Suppose the source distribution and the target distribution are denoted by $P$ and $Q$ respectively, then PE is formulated as follows:

$$\text{PE}(\theta; M, P, Q) = 1 - W_2^2(M(P; \theta), Q) / W_2^2(P, Q), \tag{1}$$

where $W_2(\cdot)$ is the Wasserstein-2 distance and $M(P; \theta)$ is notation for the mapping $M$ applied to every sample in the source, i.e. $M(P; \theta) = \{M(x; \theta) \mid x \in P\}$. Intuitively, PE quantifies how much the mapping $M(\cdot; \theta)$ reduces the distance between $P$ and $Q$. A high PE means that the explanation, $M(\cdot, \theta)$, closely matches the overall source to the overall target distribution. We can directly optimize PE using gradient descent if we use a differentiable implementation of the Wasserstein-2 distance. In our experiments, we use the GeomLoss library's (Feydy et al., 2019) differentiable Wasserstein-2 distance implementation. In what follows, when the definitions of $M, P, Q$ are clear, we will simplify $f(\theta; M, P, Q)$ as $f(\theta)$ for an arbitrary function $f$.

## 3.2 FEASIBILITY AND ROBUSTNESS METRICS

To our knowledge, PercentExplained (PE) (Equation 1) is the only metric to evaluate shift explanations. We formalize the additional metrics of feasibility and robustness, as introduced in Section 2.

**Feasibility** Counterfactual explanation literature has already defined feasibility (Poyiadzi et al., 2020). Formally, feasibility is the percentage of source samples with feasible explanations, i.e.:

$$\% \text{ Feasible} = \left[ \sum_{x \in P} a(x, M(x; \theta)) \right] / \|P\| \tag{2}$$

where $a(\cdot, \cdot)$ is a function which outputs 1 when the change from $x$ to $M(x; \theta)$ is feasible, and 0 otherwise (e.g. occupation may be feasible while age is not for the Adult dataset). Since GSE takes groups into account, we can enhance an explanation's feasibility by constructing groups using the unactionable attributes.

**Robustness** The notion of robustness is also proposed in prior work (Alvarez-Melis & Jaakkola, 2018; Agarwal et al., 2022) which evaluates variations of the explanations with respect to small perturbations to the source data distribution. To add such small perturbations to the source data distribution, $P$, we randomly perturb $\epsilon\%$ of the feature values for some pre-specified feature, e.g.,

changing the sex of 1% of the samples from male to female. The resulting perturbed source distribution is denoted as $P(\epsilon)$. We investigate the robustness of shift explanations with respect to two types of perturbations, random perturbations and worst-case perturbations. These two types of perturbations lead to two robustness metrics (denoted by $\Omega$ and $\Omega_{\text{worst}}$ respectively) which are quantified with the following formula adapted from the robustness metrics in (Alvarez-Melis & Jaakkola, 2018)):

$$\Omega(\theta; \epsilon) = \|M(P; \theta) - M(P(\epsilon); \theta(\epsilon))\|_2 / \|P - P(\epsilon)\|_2, \quad \Omega_{\text{worst}}(\theta) = \max_\epsilon \Omega(\theta, \epsilon), \tag{3}$$

in which $\theta$ and $\theta(\epsilon)$ are derived by finding a shift explanation with the source distribution as $P$, or $P(\epsilon)$, respectively. Since it is difficult to exactly solve $\Omega_{\text{worst}}$, we therefore follow (Alvarez-Melis & Jaakkola, 2018) to determine $\Omega_{\text{worst}}$ from a finite set of epsilons. Details are shown in Appendix B. Note that only $\Omega_{\text{worst}}$ is similar to measures of *model* robustness (Szegedy et al., 2014).

### 3.3 WORST-GROUP PE FOR GSE

To rectify the issues identified in Section 2 in existing shift explanations and improve feasibility and robustness, we can ideally optimize PE for all pre-specified groups such that the shift explanation preserves all groups. This ideal, however, is not applicable to finding dataset-level explanations since it requires increasing the explanation complexity by a factor of the number of groups since each group needs its own explanation. Instead, we propose Group-aware Shift Explanations (GSE) to optimize the *worst-group PE*, which leads to implicitly improving PE for *all groups* simultaneously.

Specifically, suppose the source and target are partitioned into $G$ disjoint groups, i.e., $P = \{P_1, P_2, \ldots, P_G\}$ and $Q = \{Q_1, Q_2, \ldots, Q_G\}$, in which, $P_g$ and $Q_g$ correspond to the same group, e.g., males. We can now evaluate PE on a shared group from the source and target as follows:

$$\text{PE}_g(\theta) = 1 - W_2^2(M(P_g; \theta), Q_g) / W_2^2(P_g, Q_g). \tag{4}$$

The above formula measures how much the distance between $P_g$ and $Q_g$ is reduced by the given shift explanation, $M$. The worst-group PE can then be calculated over all $G$ groups as the following:

$$\text{WG-PE}(\theta) = \min_g \text{PE}_g(\theta). \tag{5}$$

This metric indicates the reduction in distance between any pair of $P_g$ and $Q_g$, in the *worst case*. Instead of learning a shift explanation to maximize overall PE which can leave some groups with small PE, GSE learns an explanation maximizing WG-PE. Optimizing $\theta$ to maximize WG-PE can guarantee that for *every* pair of $P_g$ and $Q_g$, $\text{PE}_g(\theta)$ is not approaching 0. Intuitively, GSE penalizes explanations where subgroups have small PE even though the overall PE is large. Note that the goal of GSE is still to learn shift explanations at the dataset level rather than find explanations for each group separately. This means that the vanila $K$-cluster transport and GSE $K$-cluster transport produce explanations of the same complexity. As we will show in Section 4, both feasibility and robustness issues can also be mitigated with GSE.

### 3.4 A UNIFIED FRAMEWORK FOR GENERAL SETTINGS

In this section, we propose a generic framework which generalizes GSE from $K$-cluster transport to broad types of shift explanation methods, and from tabular data to a wide range of domains.

#### 3.4.1 GENERALIZING TO OTHER SHIFT EXPLANATION METHODS

**Generalizing the Mapping $M(x; \theta)$** Recall that shift explanations produced by $K$-cluster transport can be represented by the mapping function $M(x; \theta)$, which can be any function taking the sample $x \in P$ and the moving distance $\theta$ as input. For example, for optimal transport, $M(x; \theta) = x + \theta_x$ where the moving distance, $\theta_x$, varies between different $x$.

**Generalizing the Objective Function beyond PE** So far we have only introduced one objective function, PercentExplained (PE), for optimizing the parameters of the mapping function. Indeed, we can replace PE by any differentiable loss function, $L(\theta)$. The details of instantiating $M$ and $L$ for other shift explanation methods, e.g., optimal transport and DiCE, are in Appendix A. Note that the feasibility and robustness metrics introduced in Section 3.2 are not suitable due to their non-differentiability. Therefore, they only serve as post-hoc evaluation metrics.

We can now provide a general form of GSE for any shift explanation method decomposed as a parameterized mapping $M(\cdot; \theta)$ and an objective function $L$ for learning $\theta$. First, we extend our

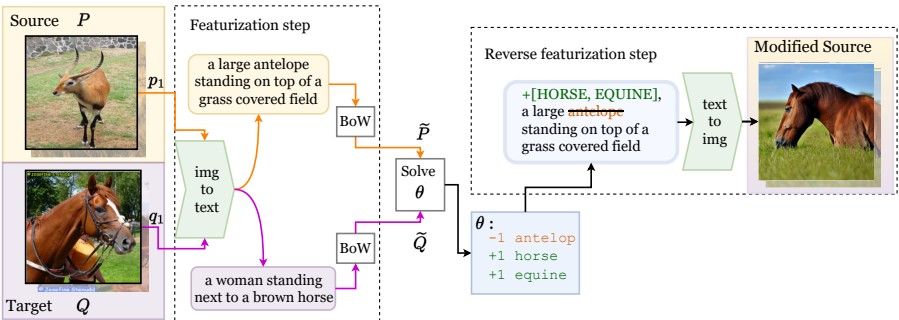

Figure 4: Pipeline of generating an interpretable shift explanation for an image using an image-to-text model for featurization and a text-to-image model for reverse featurization. Images are transformed to features in the featurization step, and we show an example of image caption-based featurization. Shift explanations are learned over interpretable features (BoW) denoted by $\widetilde{P}$ and $\widetilde{Q}$ for source and target images respectively. The modified features are converted back to images through reverse featurization, and we show an example using a text-to-image model for reverse featurization.

formulation of WG-PE in Equation 5 beyond PE by replacing PE with $1 - L$ (recall that $L$ is $1 - \text{PE}$ for $K$-cluster transport), i.e:

$$\text{WG-}L(\theta) = \min_g(\{1 - L(\theta; M, P_g, Q_g)\}_{g=1}^G) = \max_g(\{L(\theta; M, P_g, Q_g)\}_{g=1}^G) \quad (6)$$

Recall that $P_g$ and $Q_g$ represent a group of samples from the source and target respectively, belonging to the same group. We further generalize Equation 6 by using an arbitrary aggregation function $F$ in place of the $\max$ function and regularizing with the loss calculated between the whole $P$ and $Q$ to balance the optimization between the worst group and the overall distribution, i.e.:

$$\text{WG-}L(\theta; M, P, Q) = F(\{L_g(\theta; M, P_g, Q_g)\}_{g=1}^G) + \lambda \cdot L(\theta; M, P, Q) \quad (7)$$

where $\lambda$ is a hyper-parameter and $F$ is an aggregation function. The choice of $F$ and $\lambda$ for our experiments is given in Appendix D.4.

### 3.4.2 GENERALIZING TO LANGUAGE AND IMAGE DATA

Shift explanations are built upon interpretable features, for instance the age and race in the Adult dataset, but such interpretable features are not available for image and language data. Therefore, we add two additional steps in our framework. The first is a *featurization step* to extract interpretable features. Second, we add a *reverse featurization step* for converting modifications to the interpretable features back to the raw data space for producing mapped source samples.

**Generalizing to language data** For language data, the *featurization step* can leverage techniques such as Bag-of-words (BoW) and N-Gram models to produce token-level features. We can also use the embedding from pretrained language models as features which we show in Appendix H. These features for the source and target data are denoted by $\widetilde{P}$ and $\widetilde{Q}$ respectively. Then, $\widetilde{P}$ and $\widetilde{Q}$ can be integrated into $L(\theta; M, P, Q)$ and WG-$L(\theta; M, P, Q)$ for solving $\theta$. The resulting mapping function $M(\cdot; \theta)$ is in the form of removal or addition of words, or modifications to the embedding vector. Therefore, in the *reverse featurization step*, we follow the explanations to either remove or prepend words to the beginning of the input sentences, or make modifications directly in embedding space.

**Generalizing to image data** For image *featurization* and *reverse featurization*, we propose an end-to-end pipeline shown in Figure 4. This pipeline results in interpretable explanations, but we provide experiments using other featurization and reverse-featurization methods including raw image pixels, a finetuned text-to-image model, semantic image editing, and concept-based explanations (Kim et al., 2018) in Appendix H. The featurization step shown in Figure 4 starts by leveraging image-to-text models such as CLIP Interrogator (cli, 2022) to produce captions for each image. These captions are then processed in the same manner as language data to obtain interpretable features, such as BoW features, which are denoted by $\widetilde{P}$ and $\widetilde{Q}$ for the source and target respectively. We then learn shift explanations over the extracted features. Finally, the *reverse featurization step* produces images from the modified features after applying the explanation by modifying the original caption and passing it to a text-to-image model such as stable diffusion (Rombach et al., 2021).

### 3.5 THEORETICAL ANALYSIS

We theoretically analyze the existence of group irregularities in a simple 1D setting and show that our worst-group optimization method mitigates the problem.

**Theorem 1.** *Suppose $P = Bernoulli(p)$ and $Q = \alpha \cdot Bernoulli(p) + \beta$ in one dimensional setting where $p \in [0, 1]$ and $\alpha, \beta \in \mathbb{R}$ such that $\alpha + 2\beta \neq 1$. We further split the joint distribution of $P$ and $Q$ into two groups depending on the sampling results, i.e., $Group(x) = 1$ if $x \sim P$ and $x = 0$ or $x \sim Q$ and $x = \beta$, and $Group(x) = 2$ otherwise. Let $M(x; \theta_{PE}) = x + \theta_{PE}$ and $M(x; \theta_{WG}) = x + \theta_{WG}$ be two K-cluster explanations (K=1) which are solved by maximizing $PE(\theta)$ and WG-PE$(\theta)$ respectively. Then, $PE(\theta_{WG}) - WG\text{-}PE(\theta_{WG}) = 0$ for all $\alpha, \beta$, and $p$ while $PE(\theta_{PE}) - WG\text{-}PE(\theta_{PE}) > 0$ except when $p = \frac{\beta}{\alpha + 2\beta - 1}$ or $p = \frac{\beta}{1 - \alpha}$ holds.*

The takeaway is that group irregularities, or a disparity between the overall PE and worst-group PE, exist when optimizing the overall PE. We also theoretically analyze the feasibility and robustness of shift explanations in this setting in Appendix J.

## 4 EXPERIMENTS

We present our experiments for evaluating the effectiveness of GSE compared to shift explanations which ignore group structures. In what follows, we describe the experimental setup in Section 4.2, the datasets in Section 4.1, and our results in Section 4.3 and Section 4.4.

### 4.1 DATASETS

We perform experiments on tabular, language, and vision data. For tabular data, we use the Adult income (Adult) and Breast Cancer datasets (Breast) (Dua & Graff, 2017). For language data, we evaluate on the Civil Comments dataset (Borkan et al., 2019) (Civil) and Amazon review dataset (Amazon) (Ni et al., 2019). Finally, for image data we use the version of the ImageNet dataset from (Santurkar et al., 2021) (ImageNet), the FMoW dataset (Christie et al., 2018), and the iWildCam dataset (Beery et al., 2021). Appendix C provides further details on these datasets.

**Distribution shift setup** For tabular and language data, we match prior work and consider distribution shifts between class labels: shift from low to high-income for Adult, benign to malignant for Breast, toxic to non-toxic for Civil, between two sets of reviewers for Amazon, between sub-classes of "Mammal" for ImageNet, and between geographic regions for FMoW and iWildCam.

**Sub-population setup** For all datasets, we define subgroups based on intuitive notions of what subgroups exist in each dataset. We show in Appendix E that our method is useful even without group definitions. For Adult, we group samples by their sex attribute. For Breast, we group by the ratio between "cell radius" and "cell area" attributes (see Appendix D for details), leading to 3 groups. For Civil, groups are defined by samples with and without the "female" demographic feature. For Amazon, groups are defined by reviews with less than and greater than two stars. For ImageNet, groups are defined by the superclasses "rodent/gnawer" and "ungulate/hooved mammal" of the ImageNet label. For FMoW, groups are defined by geographic region, and for iWildCam, groups are defined by daytime vs. nighttime. As we show in Section 4.3, despite only a few pre-specified groups across all the datasets, the state-of-the-art shift explanations still break those group structures and lead to poor feasibility and robustness.

### 4.2 EXPERIMENTAL SETUP

For all datasets described in Section 4.1, we evaluate three shift explanation methods: $K$-cluster transport ($K$-cluster), Optimal transport (OT), and DiCE. Due to space limitations, we only include $K$-cluster transport in this section and other experiments are in Appendix C. For each method, we compare GSE to vanilla explanations. The latter ones are derived by optimizing group-free objectives such as PE in Equation 1 while the former are constructed by optimizing group-aware objectives such as WG-PE in Equation 5. We evaluate all methods along the following axes:

- PE and WG-PE (over embeddings from an ImageNet pretrained ResNet50 for image data).
- % Feasible as shown in Equation 2.
- Robustness as shown in Equation 3 by perturbing 1% of the feature values for six features.

Table 1: Comparison of PE, WG-PE, and %Feasible metrics between vanilla and GSE $K$-cluster explanations (Higher is better).

| | PE ↑ | | WG-PE ↑ | | %FEAS ↑ | |
|---|---|---|---|---|---|---|
| | VANILLA | GSE | VANILLA | GSE | VANILLA | GSE |
| BREAST | **97.56±0.00** | 96.89±0.00 | 83.42±0.00 | **93.42±0.00** | 34.43±0.00 | **35.38±0.00** |
| ADULT | **99.83±0.01** | 97.40±0.04 | 75.13±0.06 | **96.16±0.02** | 86.63±0.12 | **89.27±2.18** |
| CIVIL | 0.63±0.10 | **0.88±0.10** | 0.62±0.10 | **0.83±0.11** | 90.30±0.80 | **91.07±0.05** |
| AMAZON | -2.23±0.03 | **-1.11±0.66** | -2.41±0.03 | **-1.17±0.73** | 87.00±0.00 | 87.00±0.00 |
| IMAGENET | 18.26±1.75 | **20.07±3.45** | -8.61±4.02 | **-3.78±7.65** | 37.25±4.58 | **50.11±4.94** |
| FMoW | **18.73±0.00** | 13.02±0.00 | -17.30±0.00 | **7.46±0.00** | 50.20±0.00 | **54.55±0.00** |
| iWILDCAM | 14.60±1.29 | **15.71±0.68** | **2.39±2.07** | -1.32±0.76 | 48.69±2.28 | **71.24±2.05** |

Recall that the % Feasible and Robustness metrics are not differentiable, so we only use them for evaluation. Further details of the experimental setup are in Appendix D.

## 4.3 QUANTITATIVE RESULTS

Table 2: Comparison of Robustness and Worst-case Robustness between vanilla and GSE $K$-cluster explanations (Lower is better).

| | METHOD | ADULT | BREAST | CIVIL | AMAZON | IMAGENET | FMoW | iWILDCAM |
|---|---|---|---|---|---|---|---|---|
| ROB. ↓ | VANILLA | **1.34±0.28** | 6907.79±3577.96 | 10.10±2.26 | 18.78±11.06 | 36.56±10.43 | **8.26±0.73** | 8.48±1.51 |
| | GSE | 1.52±0.40 | **6500.83±3270.29** | **10.02±2.19** | **13.95±6.04** | **35.47±11.26** | 20.40±4.90 | **8.47±1.14** |
| WC ROB. ↓ | VANILLA | **1.74** | 16086.54 | 16.59 | 38.14 | 59.19 | **9.62** | 11.94 |
| | GSE | 2.18 | **15438.47** | **15.32** | **26.86** | **56.04** | 28.05 | **10.34** |

The main quantitative results of vanilla and GSE $K$-cluster explanations are shown in Table 1-2. We can see from Table 1 that there is often a large gap between the PE and WG-PE, showing the reality of group irregularities across datasets which was also depicted in Figure 2.

When comparing GSE explanations against vanilla explanations, GSE almost always results in a higher WG-PE (over 20% improvements on the Breast and FMoW datasets) than vanilla explanations, while minimally changing overall PE. Surprisingly, GSE improves PE as seen on the ImageNet, Civil, and iWildCam datasets. GSE also always produces more feasible explanations in comparison to vanilla explanations, and has improvements of up to 22%. This is primarily due to the fact that GSE penalizes explanations with low feasibility. Moreover, according to Table 2, GSE improves both the robustness and worst-case robustness in most cases across all datasets, by as much as 25% (see the Robustness metric for Amazon).

We see that GSE does not result in 100% WG-PE and %Feasible which is because we are using a $K$-cluster explanation with 20 clusters which is not expressive enough to entirely explain the shift in the distribution. Appendix G has results for more expressive explanation methods where we see GSE reach 100% WG-PE. Furthermore, GSE does not entirely restrict explanations to be 100% feasible which is why we also do not see perfect feasibility.

These results for image data use text-based featurization for interpretability, as shown in Figure 4. There are other featurization options and Appendix H includes additional results including explanations using raw pixel features. We see the same trends in results for the raw pixel explanations as the text explanations shown here. For full results and comparison with text features, see Appendix H.

## 4.4 QUALITATIVE RESULTS

A qualitative analysis of GSE explanations for tabular data is given in Figure 1. GSE produces an explanation which modifies age, education level, and occupation instead of changing the sex attribute (which we treat as infeasible following prior work (Poyiadzi et al., 2020)) like the vanilla explanation. For text data, we analyze an explanation from Civil with respect to its robustness in Figure 3. Finally, for image data, Figure 5 shows a shift explanation for ImageNet where we show the shift in an antelope cluster of the $K$-cluster explanation. The vanilla explanation maps antelopes to porcupines which breaks the "ungulate/hooved mammal" group, as antelopes are hooved animals while porcupines are rodents. Observing the generated examples for this cluster shows the conversion of an antelope to a porcupine yields unusual-looking results. On the other hand, GSE maps this

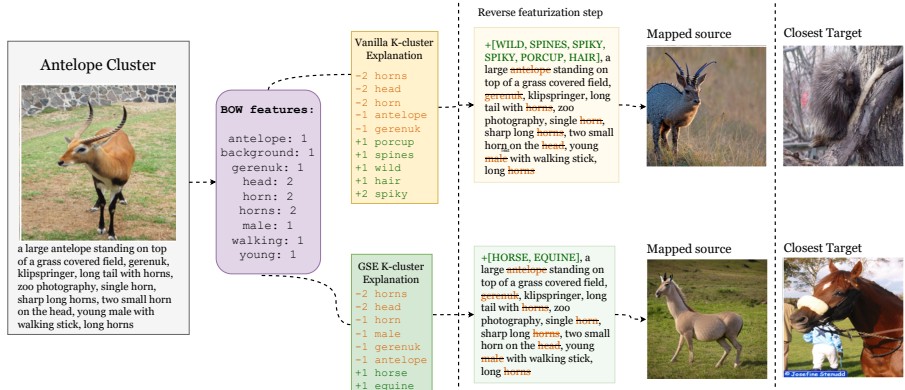

Figure 5: Vanilla vs. GSE $K$-cluster shift explanations for ImageNet sub-population shifts. Vanilla maps an antelope ("ungulate/hooved mammal" group) cluster shown on the left to porcupines shown on the top right. The explanation of "-2 horns" and "+2 spiky" means that two occurrences of the word "horns" should be removed and "spiky" should be added twice to the caption. In contrast, GSE preserves 'ungulate/hooved mammal' structure, mapping antelope to horses (bottom right). Source images for both techniques are generated using reverse featurization (Section 4).

cluster of antelopes to horses which preserves the groups since horses are also hooved animals. The resulting generated images from this explanation are also clearly images of horses which explains why GSE has higher feasibility than vanilla explanations for image data.

## 5 RELATED WORK

**Explaining distribution shift.** Kulinski & Inouye (2023) proposes three different mappings of varying levels of interpretability and expressiveness as shift explanations. Finding counterfactual explanations to explain model behavior (Mothilal et al., 2020) is a related problem, where such explanations represent the minimal perturbation which changes a model's prediction on a given sample (Wachter et al., 2017; Chang et al.; Rathi, 2019). Although not originally created to explain distribution shift, we adapt these methods to our setting (see Appendix A for details).

**Worst group robustness.** Improving model robustness over sub-populations using group information is extensively studied. Here, the main goal is to minimize the loss on the worst performing sub-population which often becomes a form of distributionally robust optimization (DRO) (Ben-Tal et al., 2013). The problem of improving model robustness or accuracy on subgroups can be addressed through applications of DRO over subgroups (Sagawa et al., 2019; Zhang et al.), re-weighting sub-populations (Liu et al., 2021; Byrd & Lipton, 2019), or performing data augmentation on the worst group (Goel et al.). Rather than focus on improving model robustness, our focus is on finding explanations that preserve group structures.

**Domain generalization and adaptation.** Common solutions for dealing with distribution shift in regards to a model include *domain generalization* and *domain adaptation*. Unlike these methods, our setting is independent of a model. We survey these methods in detail in Appendix I.

## 6 CONCLUSION AND FUTURE WORK

We identified a problem with all existing approaches for explaining distribution shift: the blindness to group structures. Taking group structures into account, we developed a generic framework that unifies existing solutions for explaining distribution shift and allows us to enhance them with group awareness. These improved explanations for distribution shift can preserve group structures, as well as improve feasibility and robustness. We empirically demonstrated these properties through extensive experiments on tabular, language, and image settings.

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

## A   ADDITIONAL FRAMEWORK INSTANTIATIONS

### A.1   OPTIMAL TRANSPORT (OT)

Similar to $K$-cluster Transport ($K$-cluster) (Kulinski & Inouye, 2023), Optimal Transport (OT) finds the moving distance $\theta(x)$ for shift explanations directly. In the next two sub-sections, we discuss how to instantiate $M(x; \theta)$ and $L(\theta; M, P, Q)$ for OT within our framework.

**Mapping function for OT.** In OT, the mapping is almost the same as that for $K$-cluster except that the moving distance $\theta$ now depends on each individual sample, $x$, from the source. Therefore, the counterfactual mapping $M(x; \theta)$ can be written as $M(x_i; \theta_i) = x_i + \theta_i$ for every $x_i \in P$.

**Objective function for OT.** The objective function for OT is exactly the same as that for $K$-cluster which is the PE metric. The optimization now results in learning $\theta = \{\theta_1, \ldots, \theta_{|P|}\}$, or a separate moving distance for every source sample such that the PercentExplained is maximized.

### A.2   DICE

For vanilla counterfactual explanation methods such as DiCE, model behavior for a given sample $x$ is explained. To construct such explanations, these methods perform counterfactual modifications to $x$ such that the model prediction changes. We adapt these methods to construct a surrogate shift explanation by finding counterfactual examples for models that classify between source and target distributions. In this subsection, we investigate how general methods for finding counterfactual examples can be adapted to fit within our framework. We take DiCE as an example to describe how to instantiate $M(x; \theta)$ and $L(\theta; M, P, Q)$ for these methods.

**Mapping function for DiCE.** The counterfactual examples produced by DiCE depend on a given model (parameterized by $\theta$). As a consequence, the mapping function for DiCE, $M(x, \theta)$, is represented as $M(x, \theta) = x + f(x; \theta)$. Let $h$ denote the fixed model which classifies between the source and target data. The moving distance, $f(x; \theta)$, used in the counterfactual explanation relies on this model, $h$, that DiCE is used to explain.

**Objective function for DiCE.** As indicated above, it is essential to obtain the parameter $\theta$ to learn the shift explanation. Since the model, $h$, discriminates between the source data, $P$, and the target data, $Q$, we optimize the following objective function for DiCE, in which all source samples and target samples are labeled as 0 and 1 respectively:

$$\arg\min_{\theta} L_{\text{DiCE}}(\theta; M, P, Q)$$
$$= \arg\min_{\theta} \sum_{x,y \in D} \ell(h(x; \theta), y). \tag{8}$$

In the above formula, the loss $\ell(\cdot)$ represents the Cross Entropy loss and $h$ denotes the model which classifies between the source and target data $D = \{(x, 0) : x \in P\} \cup \{(x, 1) : x \in Q\}$. Note that the above loss function is an instantiation of the abstract objective function, $L$, used in Equation

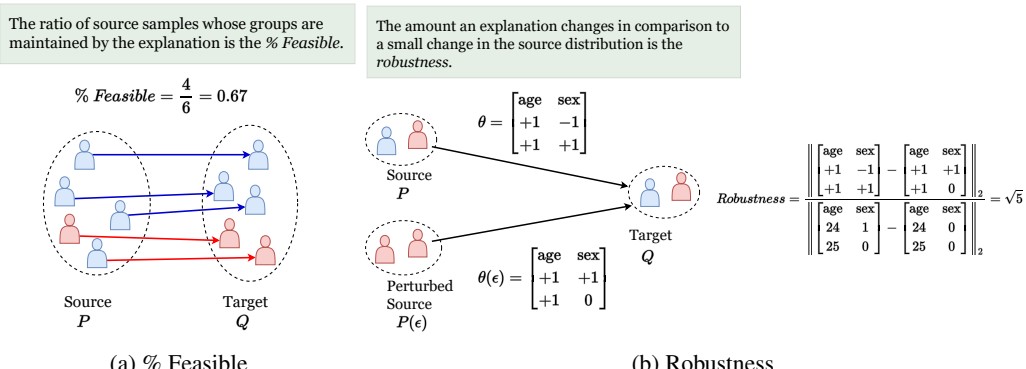

(a) % Feasible                    (b) Robustness

Figure 6: Visualizations of feasibility and robustness. % Feasible is shown in Figure 6a and it measures the percent of samples which are mapped by $G(x; \theta)$ to a sample with the same group as the original sample. Robustness is shown in Figure 6b and it measures how small perturbations on the source data distribution change shift explanation.

equation 7. This optimization leads to learning a $\theta$ which is the model parameter for the classifier between the source and target. Once we have learned the model parameter for the model $h$ to be explained with DiCE, we derive the moving distance as

$$
\begin{aligned}
f(x; \theta) &= \mathrm{argmin}_{\delta_x} \mathrm{dist}(x, x + \delta_x) \\
s.t.\ & h(x + \delta_x; \theta) = 1.
\end{aligned}
\tag{9}
$$

For any $x \in P$ the moving distance $f(x; \theta)$ is found such that it is a minimal change to $x$ which results in the previously learned classifier, $h$, classifying the modified sample as a target sample.

## B    DETAILS ON FEASIBILITY AND ROBUSTNESS

Feasibility and robustness are defined in Equation 2 and 3 respectively, but here we give a visual example of each. A concrete example for calculating feasibility is shown in Figure 6a. The source cluster of four males and one female becomes three males and two females from the mapping, so feasibility is $\frac{4}{6} = 0.667$.

Similarly, we calculate robustness for an example in Figure 6b. Suppose there are two clusters in the source distribution and the target distribution respectively, and each cluster consists of a single sample. After applying $K$-cluster transport, the moving distance $\theta$ from the source to the target can be interpreted as "increasing the age by 1 and flipping the sex attribute". After perturbing the sex attribute of one source sample from 1 to 0, the magnitude of changes on the source data distribution is $\|P - P(\epsilon)\|_2 = 1$. This produces a new moving distance $\theta(\epsilon)$, which is interpreted as "increasing the age by 1 and only flipping the sex attribute of the first source sample". By leveraging Equation equation 3, the Robustness measure $\Omega$ for this example is $\frac{\|M(P; \theta) - M(P(\epsilon); \theta(\epsilon))\|_2}{\|P - P(\epsilon)\|_2} \approx \frac{\|\theta - \theta(\epsilon)\|_2}{\|P - P(\epsilon)\|_2} = \sqrt{5}$.

The details for how we produce a perturbation, calculate worst-case robustness, and perform the robustness experiment are given in Appendix D.5.

## C    DATASETS

The tabular, language, and image datasets that we use in the experiments are described in this section.

### C.1    TABULAR DATA

**Dataset overview.** The Adult dataset and the Breast Cancer dataset are standard tabular datasets from the UCI Machine Learning Repository (Dua & Graff, 2017). The Adult dataset consists of 48,842 samples with categorical and integer features from census data. The typical task is to predict whether income exceeds $50K per year. The Breast Cancer dataset contains 569 samples with 10

real-valued features relating to an imaged cell. This dataset is similarly used for binary classification between the classes of benign and malignant tumors.

**Distribution shift setup.** For both the Adult and Breast Cancer datasets, we match the setup by (Kulinski & Inouye, 2023) and consider distribution shift between the different class labels: above 50k and below 50K for Adult, and benign and malignant for Breast Cancer.

**Sub-population setup.** For the Adult dataset, we use the existing demographic feature of "male" to define two groups. For the Breast Cancer data, we define groups by thresholding on a new attribute which is calculated by using "cell radius" and "cell area" attributes (see Appendix D for details). This leads to 3 groups in total.

## C.2 LANGUAGE DATA

**Dataset overview.** The Civil Comments dataset (Borkan et al., 2019) and the Amazon review dataset (Ni et al., 2019) are used for our language application. The Civil Comments dataset targets predicting the toxicity of up to 2 million public comments and it additionally contains annotations of demographic categories including gender, race, and religion of the authors of each comment. The Amazon review dataset is used for sentiment classification where there is a distribution shift between the subpopulations of reviewers. These datasets are part of the WILDS (Koh et al., 2021) distribution shift benchmarks and they are used to benchmark subpopulation shift. Subpopulation shift occurs when the proportions of samples from different demographic categories changes between the source and target.

**Distribution shift, sub-population and featurization setup.** For the Civil Comments dataset, we build a distribution shift setting by splitting it into toxic and non-toxic text as the source and target respectively as done by (Kulinski & Inouye, 2023). After balancing the size of this split, there are 4,437 samples in each of the source and target. The groups are defined by samples with and without the "female" demographic feature. For the Amazon review dataset, we use the default split from the WILDS benchmark which splits the data into two sets of reviews with reviews from different reviewers between the two sets, and we define two groups based on the year the review was written. After balancing the size of the source and target, there are 4,910 samples in each.

The interpretable features for this data are defined by the bag-of-words representation for each sample. We limit the bag-of-words to 50 words which we find helps to avoid model overfitting when using DiCE.

## C.3 IMAGE DATA

**Dataset overview.** For image data, we use BREEDS (Santurkar et al., 2021), FMoW (Christie et al., 2018), and iWildCam (Beery et al., 2020). The BREEDS dataset is a collection of ImageNet (Deng et al., 2009) subsets created using the wordnet class hierarchy for sub-population shift studies. For our experiments, we take a subset of the ImageNet validation set based on a subpopulation shift between hooved mammals and rodents. The FMoW dataset consists of over 1 million satellite images labelled with one of 62 building or land use categories. This dataset is used to study domain generalization and subpopulation shift since it additionally provides geographic region and time attributes which can be used for creating subsets exhibiting different distribution shifts. Finally, the iWildCam dataset consists of camera trap images taken in different geographic regions. This dataset is used to study subpopulation shift because the distributions of animals shifts between regions.

**Distribution shift setup.** In BREEDS, We start at the subtree under "mammal" in ImageNet's wordnet hierarchy and select three ImageNet classes under the superclass "rodent/gnawer" and three classes under the superclass "ungulate/hooved mammal" for both the source and target. These three classes for each superclass are chosen in an adversarial way according to (Santurkar et al., 2021) to increase the level of subpopulation shift. In total, this subset consists of 298 samples in each of the source and target. In our experiments on FMoW, we subset to the first three land use / building classes and construct the source distribution from samples taken before 2012 and the target from samples taken after 2012 which results in 253 samples for the source and 279 samples for the target. In our experiments on iWildCam, we take the source distribution as images from a single region and the target distribution as images from a separate region. This results in 204 samples for the source and target each.

Table 3: Learning rates for all experiments

| Dataset | OT | $K$-cluster | DiCE |
|---------|-----|-------------|------|
| Adult | 0.1 | 0.5 | 0.1 |
| Breast | 0.1 | 5 | 0.5 |
| Civil | 0.1 | 0.5 | 0.5 |
| ImageNet | 0.5 | 1.0 | 0.5 |
| FMoW | 0.1 | 0.5 | 0.005 |
| iWildCam | 1.0 | 1.0 | 0.5 |
| Amazon | 0.5 | 0.5 | 0.5 |

**Featurization and sub-population setup.** As described in Section 3.4.2, features for BREEDS and iWildCam are extracted by using an img-to-text model and then treating the caption as a bag-of-words representation. We use a total of 50 words in the bag-of-words as features. Groups are defined for BREEDS by the superclasses "rodent/gnawer" and "ungulate/hooved mammal" to encourage an explanation which does not map rodents to hooved mammals. This grouping allows us to define an infeasible explanation as one which maps rodents to hooved mammals or vice versa. For the FMoW dataset, we featurize based on raw pixel values since the text-to-image model does not perform well on generating satellite images. To construct groups, we use the provided geographic region attribute to define groups since we want our distribution shifts to respect geographic boundaries. For the iWildCam dataset, we define groups by the daytime and nighttime attribute of the images since we don't expect that animals in one region will become nocturnal in another region.

Note that for datasets such as Civil Comments, iWildCam, and FMoW, the groups are determined by extra annotations which are not available after mapping the source samples by the shift explanations. To determine the group assignments of a mapped source sample, we leverage the group annotation of the mapped sample's closest target sample as an approximated annotation.

## D DATASETS AND HYPERPARAMETERS FOR EXPERIMENTS

### D.1 TABULAR DATA

All categorical features in the Adult data are one-hot encoded resulting in a total of 35 features. We balance the size of both source and target distribution which results in a total of 15,682 samples for the Adult data and 424 samples for the Breast dataset.

The new meta-feature that is used for grouping the Breast dataset is calculated by the expression

$$\frac{\text{Avg. cell radius}^2}{\text{Avg. cell area}},$$

and then we group the data by thresholding on this meta-feature. To find a good threshold, we compute the meta-feature for the entire source and target dataset and get the first and third quartiles. Thus, we create three groups: samples with meta-feature value below the first quartile, between the first and third quartile, or above the third quartile.

When learning vanilla and GSE explanations, we use the same hyperparameters between both. For all methods, we optimize for 200 iterations and list all the specific hyperparameters in Table 3. For DiCE, we use a neural network with a single hidden layer of size 16 as the source vs. target discriminator in all experiments. This network is trained for 500 iterations, with a weight decay of 0.0001. For GSE DiCE, we train the neural network using group DRO (Sagawa* et al., 2020) with the same hyperparameters used for the vanilla training.

### D.2 LANGUAGE DATA

For the $K$-cluster experiments, we use 4 clusters and optimize for 200 iterations using a learning rate of 20. For OT explanations, we optimize for 200 iterations using a learning rate of 0.1. Finally, for the DiCE explanations we first train a logistic regression classifier for classifying the source and target samples using 1000 epochs with learning rate of 0.5 and weight decay of 0.0001. For GSE with DiCE, we train this logistic regression classifier using group DRO with the same hyperparameters used in the regular training procedure.

### D.3 IMAGE DATA

For $K$-cluster explanations, we use 5 clusters and optimize for 100 iterations using a learning rate of 150.0. For OT explanations, we optimize for 100 iterations using a learning rate of 0.5. Finally, for the DiCE explanations we first train a logistic regression classifier for classifying the source and target samples using 100 epochs with learning rate of 0.1 and weight decay of 0.0001, and we use group DRO to train this classifier for GSE.

### D.4 FRAMEWORK HYPERPARAMETERS

For all experiments, we use the sum function $F(X) = \sum_{x \in X} x$ for the function $F$ in Equation 7. We also experimented with group DRO loss (Sagawa* et al., 2020) and $F(X) = \max X$ with $\lambda = 0.1$. Note that for the summation $F(X)$, it is only applicable to the loss function $L$ which does not preserve the addition operation over groups, such as PE. Otherwise, Equation 7 could be rewritten as $\min_\theta ((1 + \lambda) \cdot L(\theta; M, P, Q))$, which is not a group-aware loss.

### D.5 ROBUSTNESS EXPERIMENT

To compute the robustness metric, we use a random small perturbation to the source distribution. To create this perturbation, we randomly select 75% of the features and perturb 1% of the feature values for each of these features. The manner in which we perturb this 1% of the feature values depends on the type of the feature. For real valued features, we find the standard deviation of the feature value for the current feature we are perturbing and we randomly either add or subtract $0.05 \cdot$ stdev to 1% of the feature values. For integer features, we randomly either add or subtract 1 to 1% of the feature values. Finally, for boolean features, we randomly either flip the label of 1% of the True feature values or 1% of the False features values. For categorical features, we first convert the categories to integers such that each category is given an integer from 0 to $K$-1 where $K$ is the number of categories. This allows us to generate a perturbation for categorical features in the same way as for integer features.

We use the same hyperparameters as above for learning each shift explanation on the perturbed distribution. To speed up the experiments, we first train the shift explanation on the original source distribution and then initialize the parameters of the shift distribution with the parameters learned from the original source distribution when learning the shift explanation for the perturbed distribution.

For computing the robustness metric, we use three random perturbations as described above and average the robustness over the three runs. To compute worst-case robustness, we calculate robustness from 100 random perturbations and take the worst (highest) value of robustness from the 100 trials. Since each calculation of robustness requires learning a shift explanation using the vanilla method and GSE, this experiment is time consuming, so we don't report error bars for the worst-case robustness.

### D.6 COMPUTE DETAILS

For all experiments, we use a local server with four Nvidia 2080 Ti GPUs and 80 Intel Xeon Gold 6248 CPUs. Each experiment required around 2 GB of GPU memory.

## E EXPERIMENTS WITHOUT GROUP LABELS

It is possible that group labels are not always available for a dataset, but we can still use either pretrained models to extract attributes to use for defining groups or use unsupervised methods for grouping the data. We perform an experiment on the language data to show that our group-aware method is still applicable even without group supervision.

To get groups for the language data, we cluster the sentence embeddings of our source and target data. The sentence embeddings are from a state-of-the-art sentence embedding model, `all-mpnet-v2`[2], and we use K-means clustering with 10 clusters to get 10 groups for the source

---
[2]https://huggingface.co/sentence-transformers/all-mpnet-base-v2

Table 4: Comparison of distribution shift explanation methods on Civil without groups given.

| Method | PE | WG-PE | % Feas. | Robustness | WC Robustness |
|---|---|---|---|---|---|
| DiCE | $1.08 \pm 0.1$ | $-5.84 \pm 0.5$ | $54.50 \pm 0.82$ | $7.82 \pm 0.01$ | 7.93 |
| GSE DiCE | $\mathbf{14.32 \pm 0.9}$ | $\mathbf{7.52 \pm 0.5}$ | $\mathbf{56.33 \pm 0.85}$ | $7.71 \pm 0.08$ | **7.92** |
| $K$-cluster | $5.19 \pm 1.75$ | $2.64 \pm 0.34$ | $66.00 \pm 0.71$ | $\mathbf{3.00 \pm 0.20}$ | 4.60 |
| GSE $K$-cluster | $\mathbf{5.72 \pm 0.88}$ | $\mathbf{3.79 \pm 0.27}$ | $\mathbf{67.00 \pm 0.41}$ | $3.02 \pm 0.05$ | **3.21** |
| OT | $\mathbf{99.89 \pm 0.00}$ | $63.07 \pm 2.97$ | $55.17 \pm 4.11$ | $\mathbf{1.00 \pm 0.02}$ | **1.05** |
| GSE OT | $98.34 \pm 0.28$ | $\mathbf{93.48 \pm 0.26}$ | $\mathbf{84.67 \pm 0.24}$ | $1.06 \pm 0.03$ | 1.14 |

and target. Experimental results are shown in Table 4, and we see the same trends as for the experiments with specified groups. In particular, our group-aware explanation always results in higher worst-group PE and % Feasible than the regular explanation. The most significant improvement in WG-PE is seen for the OT explanation with a change from 63.07% to 93.48%. Interestingly, we also see that our group-aware explanation has slightly improved overall PE over the vanilla DiCE and $K$-cluster explanations.

## F  MOTIVATING EXAMPLE DETAILS

### F.1  FULL RESULTS FOR MOTIVATING EXAMPLE

The full results for the explanations shown in the motivating example in Figure 1 and Figure 3.

Table 5: Full results for learning a $K$-cluster explanation for low to high income shift in the Adult dataset. Groups are defined by Black and White racial groups.

| Method | PE | White PE | Black PE | % Feas. | Robustness | WC Rob. |
|---|---|---|---|---|---|---|
| $K$-cluster | $\mathbf{86.29 \pm 0.09}$ | $\mathbf{87.62 \pm 0.06}$ | $51.77 \pm 1.60$ | $24.17 \pm 2.57$ | $3.69 \pm 1.98$ | 6.49 |
| GSE $K$-cluster | $70.22 \pm 0.65$ | $68.72 \pm 0.80$ | $\mathbf{67.39 \pm 0.05}$ | $\mathbf{100.00 \pm 0.00}$ | $\mathbf{1.93 \pm 0.95}$ | **3.27** |

### F.2  ADDITIONAL EXAMPLE

We provide another motivating example to show that different choices of infeasible features is possible. Figure 7 can be used in place of Figure 1 and Figure 8a in place of Figure 3a. The choice of which features are infeasible is entirely dependent on the user of the shift explanation and their goals.

## G  RESULTS FOR OT AND DiCE SHIFT EXPLANATIONS

The full results for tabular data, Civil Comments, and ImageNet are given in Table 6, 7, and 8 respectively. With DiCE and OT shift explanations, we see the same trends as previously mentioned in relation to $K$-cluster explanations. In particular, WG-PE is always improved by GSE, and feasibility and robustness are improved in most cases.

## H  IMAGE AND TEXT FEATURIZATION ABLATION

Table 9 and Table 10 include the results of an ablation on the featurization method used in the experiments. For the BREEDS dataset, we experiment with four different featurization techniques: using raw image pixels, using the Stable Diffusion (Rombach et al., 2021) text-to-image model finetuned on the BREEDS source and target dataset using LoRA (Hu et al., 2022), using a recent semantic image editing technique LEDITS (Tsaban & Passos, 2023), and using embeddings from a state-of-the-art classification model and interpreting them using concepts (Kim et al., 2018). We extract embeddings using ViT-Huge (Dosovitskiy et al., 2021) and learn concepts using ACE (Ghorbani et al., 2019) by constructing roughly 32 patches per image for a sample of 200 images from the source and target and then using KMeans to get 100 clusters in the ViT-Huge embedding space.

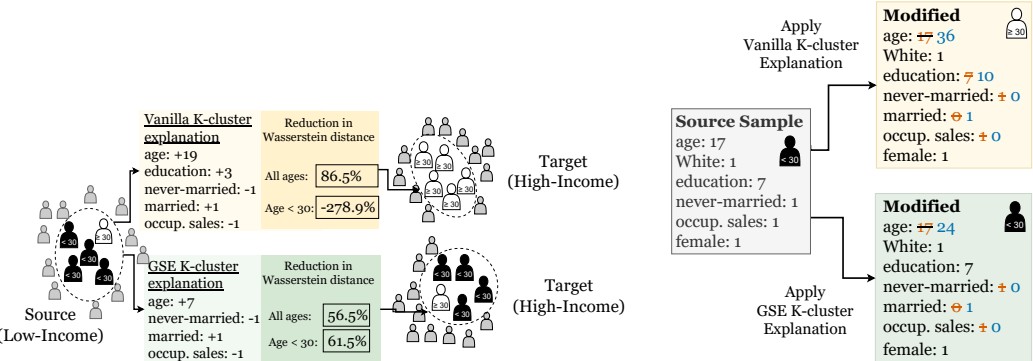

(a) Cluster level shift explanation.    (b) Instance level shift explanation.

Figure 7: (a) shows an example shift explanation from a low-income population to a high-income population from the Adult dataset using two different methods: Vanilla $K$-cluster transport and our GSE $K$-cluster transport. The shift explanation produced by the Vanilla method explains the shift by increasing age by 19 years, increasing total education by 3 years, making the cluster married, and making the cluster no longer work in sales. On the other hand, GSE generates an explanation that better preserves the subpopulations of people less than 30 and those at least 30 years old depicted as black and white figures respectively, by modifying the age feature by only seven years. The change of +3 to education means increasing the maximum education level in grades by three years. (b) shows the instance-level explanation. This explanation changes the sample's age from 17 to 36 whereas our method only increases age to 24.

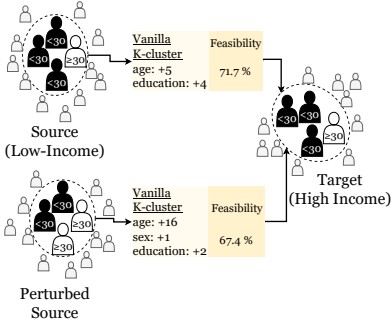

(a) Adult dataset.

Figure 8: Examples of poor robustness of an explanation. Even if an explanation is feasible (top), small perturbations to the source distribution can make it become infeasible (bottom).

Once we learn a shift explanation in the ViT-Huge embedding space, we interpret the explanation in terms of the 100 concepts and visualize the explanation for one source cluster in Figure 9.

For the Civil dataset, we experiment with two different featurization techniques: using topic modeling to extract features and perform reverse featurization and using text embeddings from a language model and an embedding inversion technique (Morris et al., 2023) to interpret them. Specifically, for topic modelling, we follow the topic modeling technique from Dieng et al. (2020) to construct a topic model over the Civil source and target data. For the explanation in embedding space, we use text-embedding-ada-002 (Greene et al., 2022) to get text embeddings for all samples in the source and target datasets. After learning a shift explanation, we decode the shifted source centroids to text using vec2text (Morris et al., 2023). Figure 10 shows an example explanation learned on text embeddings where we visualize a decoded centroid and the mapped centroid from the Vanilla $K$-cluster method compared to our $K$-cluster method.

In general, these results show the same trends as our previous results which empirically supports our claim that our framework is not dependent on a given featurization technique. For BREEDS, even if we choose no featurization (raw pixels), we still see that GSE reduces group irregularities and improves feasibility. Interestingly, there is not always a clear tradeoff between interpretability

Table 6: Comparison of distribution shift explanation methods on tabular datasets.

(a) Adult data

| Method | PE | WG-PE | % Feas | Robustness | WC Robustness |
|---|---|---|---|---|---|
| Vanilla DiCE | $2.25 \pm 0.29$ | $2.25 \pm 0.24$ | $100.0 \pm 0.00$ | $23.74 \pm 4.05$ | 41.58 |
| GSE DiCE | $\mathbf{26.02 \pm 3.00}$ | $\mathbf{21.69 \pm 4.77}$ | $100.0 \pm 1.52$ | $\mathbf{22.34 \pm 1.29}$ | $\mathbf{34.56}$ |
| Vanilla OT | $95.88 \pm 0.08$ | $80.39 \pm 0.16$ | $84.87 \pm 0.52$ | $\mathbf{0.77 \pm 0.16}$ | $\mathbf{1.22}$ |
| GSE OT | $\mathbf{96.07 \pm 0.03}$ | $\mathbf{90.91 \pm 0.17}$ | $\mathbf{91.7 \pm 4.67}$ | $0.79 \pm 0.17$ | 1.23 |

(b) Breast data

| Method | PE | WG-PE | % Feas | Robustness | WC Robustness |
|---|---|---|---|---|---|
| Vanilla DiCE | $29.6 \pm 2.43$ | $25.16 \pm 1.00$ | $25.94 \pm 0.00$ | $\mathbf{4908.99 \pm 3886.07}$ | 15504.20 |
| GSE DiCE | $\mathbf{38.21 \pm 1.58}$ | $\mathbf{33.48 \pm 0.40}$ | $\mathbf{27.20 \pm 1.46}$ | $5001.72 \pm 2924.73$ | $\mathbf{11334.15}$ |
| Vanilla OT | $99.37 \pm 0.03$ | $84.10 \pm 0.03$ | $39.62 \pm 0.77$ | $89.88 \pm 24.95$ | 112.00 |
| GSE OT | $\mathbf{99.87 \pm 0.00}$ | $\mathbf{99.37 \pm 0.00}$ | $\mathbf{93.87 \pm 0.00}$ | $\mathbf{45.51 \pm 8.20}$ | $\mathbf{56.44}$ |

Table 7: Comparison of distribution shift explanation methods on language data.

(a) Civil Comments

| Method | PE | WG-PE | % Feas. | Robustness | WC Robustness |
|---|---|---|---|---|---|
| DiCE | $2.75 \pm 0.19$ | $1.11 \pm 0.30$ | $63.33 \pm 1.25$ | $5.28 \pm 1.72$ | 6.75 |
| GSE DiCE | $\mathbf{19.29 \pm 0.80}$ | $\mathbf{15.12 \pm 2.47}$ | $\mathbf{64.67 \pm 0.62}$ | $\mathbf{1.72 \pm 0.06}$ | $\mathbf{3.40}$ |
| OT | $3.03 \pm 0.07$ | $2.51 \pm 0.16$ | $88.10 \pm 0.14$ | $4.45 \pm 0.79$ | 5.89 |
| GSE OT | $\mathbf{3.24 \pm 0.14}$ | $\mathbf{3.17 \pm 0.12}$ | $\mathbf{95.07 \pm 0.05}$ | $\mathbf{4.24 \pm 0.74}$ | $\mathbf{5.56}$ |

(b) Amazon Review

| Method | PE | WG-PE | % Feas. | Robustness | WC Robustness |
|---|---|---|---|---|---|
| DiCE | $-17.02 \pm 0.19$ | $-17.02 \pm 0.19$ | $79.0 \pm 0.08$ | $967.04 \pm 161.30$ | – |
| GSE DiCE | $\mathbf{-16.79 \pm 0.18}$ | $\mathbf{-16.79 \pm 0.18}$ | $79.00 \pm 0.05$ | $\mathbf{897.17 \pm 127.22}$ | – |
| OT | $\mathbf{-1.10 \pm 0.02}$ | $\mathbf{-1.21 \pm 0.07}$ | $79.03 \pm 0.25$ | $\mathbf{25.31 \pm 5.03}$ | $\mathbf{36.11}$ |
| GSE | $-1.25 \pm 0.20$ | $-1.25 \pm 0.20$ | $\mathbf{88.10 \pm 0.36}$ | $29.60 \pm 4.80$ | 36.50 |

of the features used and the resulting PE of the explanation. For instance, comparing $K$-cluster with raw pixel features for BREEDS to $K$-cluster with text features, we see that pixel features result in slightly lower overall PE than text features. This is because the text-based explanation can add, for example, different horses based on the other features in an image while the pixel explanation must add the same horse to each sample in a cluster.

We emphasize that extracting interpretable features is a separate problem to what this paper studies, but our methods and framework are independent of feature choice, so as new featurization techniques are developed, they can be used with our method.

Some of the worst-case robustness results in Table 9 and Table 10 are marked as "–" since the experiment took too long to run, and we omit DiCE results from the Concepts and Embedding featurization since DiCE did not scale to data with a dimension of 1536 and 768 respectively.

We also visualize the concept featurization explanations and the language embedding featurization explantions. The explanation learned over image embeddings is visualized in Figure 9 in terms of concepts. We learned concepts by taking patches from 200 samples from the source and target and then using KMeans clustering where cluster centroids were treated as concepts. This is equivalent to ACE (Ghorbani et al., 2019), a common concept extraction method.

Table 8: Comparison of distribution shift explanation methods on image data.

(a) ImageNet

| Method | PE | WG-PE | % Feas. | Robustness | WC Robustness |
|---|---|---|---|---|---|
| DiCE | -1.09 ± 1.54 | -17.25 ± 2.55 | **50.39 ± 0.42** | **5.08 ± 0.36** | 16.24 |
| GSE DiCE | **0.19 ± 1.63** | **-15.27 ± 3.08** | 49.94 ± 0.32 | 6.39 ± 0.64 | **15.73** |
| OT | 7.18 ± 1.04 | -17.30 ± 2.74 | 36.12 ± 0.55 | 18.77 ± 2.56 | 24.33 |
| GSE OT | **12.81 ± 1.34** | **-14.70 ± 2.50** | **48.16 ± 0.72** | **7.76 ± 1.73** | **22.79** |

(b) iWildCam

| Method | PE | WG-PE | % Feas. | Robustness | WC Robustness |
|---|---|---|---|---|---|
| DiCE | 6.26±1.18 | -1.02±1.67 | 67.16±0.40 | 4.37±0.59 | – |
| GSE DiCE | **7.61±0.22** | **0.92±0.42** | **67.97±0.46** | **3.96±1.01** | – |
| OT | 14.36±0.10 | **3.59±0.77** | 66.67±0.47 | 2.89±1.36 | 7.46 |
| GSE OT | **18.64±0.26** | -1.77±0.58 | **100±0.0** | **2.72±1.32** | **7.27** |

(c) FMoW

| Method | PE | WG-PE | % Feas. | Robustness | WC Robustness |
|---|---|---|---|---|---|
| DiCE | -17.04±0.09 | -17.04±0.09 | 51.52±0.19 | **706.15±22.32** | – |
| GSE DiCE | **-16.57±0.44** | **-16.57±0.44** | **51.65±0.19** | 730.96±25.03 | – |
| OT | **98.78±1.42** | -8.34±0.0 | 47.43±0.0 | 5.24±0.77 | 6.82 |
| GSE OT | 89.72±0.0 | **85.34±0.0** | **97.63±0.0** | **4.75±0.79** | **6.26** |

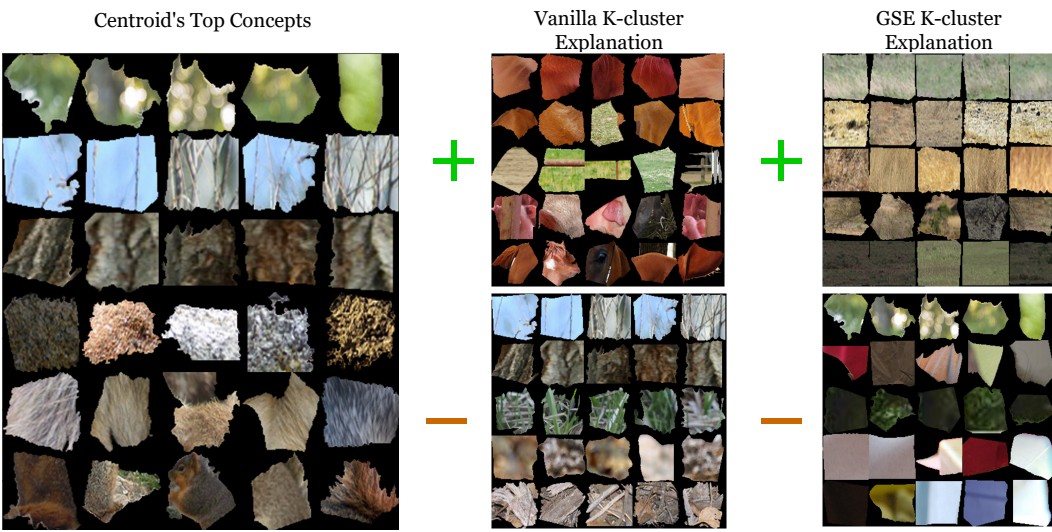

Figure 9: Concept visualization of a Vanilla and GSE $K$-cluster shift explanation learned for sub-population shift in the BREEDS dataset using embedding space featurization. On the left is are the top concepts for a centroid of the source data where patches from the same concept are in a row and different rows are different concepts. This cluster contains squirrel (rodent), fur, and different tree and grass concepts. The Vanilla explanation adds different concepts associated with pigs and horses (hooved mammals) thus breaking the rodent group of this cluster. The GSE explanation adds different types of grass concepts and removes highly saturated single color concepts and blurred background concepts.

# I  ADDITIONAL RELATED WORK

**Domain generalization and adaptation.** Common solutions for dealing with distribution shift include *domain generalization* and *domain adaptation*. Domain generalization assumes that the target

Table 9: BREEDS featurization ablation

(a) Raw pixels

| Method | PE | WG-PE | % Feas. | Robustness | WC Robustness |
|---|---|---|---|---|---|
| DiCE | -4.22±3.00 | -4.87±3.46 | 54.70±0.55 | **1222.97±19.99** | – |
| GSE DiCE | **-2.32±3.30** | **-2.67±3.79** | **55.26±0.16** | 1224.29±51.85 | – |
| OT | 100.0±0.0 | 36.65±0.17 | 51.45±0.32 | 581.71±8.97 | – |
| GSE OT | 100.0±0.0 | **100.0±0.0** | **100.0±0.0** | **1.0±0.0** | – |
| $K$-cluster | **13.25±0.05** | 9.42±0.06 | 53.91±0.16 | 104.59±72.67 | – |
| GSE $K$-cluster | 12.72±0.01 | **12.72±0.01** | **57.16±0.32** | **81.56±109.53** | – |

(b) Finetuned Stable Diffusion

| Method | PE | WG-PE | % Feas. | Robustness | WC Robustness |
|---|---|---|---|---|---|
| DiCE | **4.43±0.90** | **-8.58±1.83** | **53.69±1.45** | 12.52±3.24 | – |
| GSE DiCE | 3.88±1.13 | -10.45±2.34 | 51.57±0.84 | **9.96±2.89** | – |
| OT | 18.18±1.19 | -9.80±0.63 | 40.83±0.16 | **20.61±6.78** | – |
| GSE OT | **20.48±0.28** | **-8.24±0.67** | **55.82±1.58** | 21.28±7.98 | – |
| $K$-cluster | 7.03±0.57 | -14.83±3.01 | 25.50±2.96 | **29.58±1.45** | – |
| GSE $K$-cluster | **7.07±0.61** | **-13.09±0.84** | **30.09±1.78** | 31.50±2.00 | – |

(c) LEDITS

| Method | PE | WG-PE | % Feas. | Robustness | WC Robustness |
|---|---|---|---|---|---|
| DiCE | -0.29±0.14 | -7.86±0.54 | 25.95±0.42 | **6.53±0.83** | – |
| GSE DiCE | **32.32±0.52** | **-2.52±0.70** | **42.51±0.96** | 8.33±1.81 | – |
| OT | 2.39±0.90 | -8.54±0.75 | 18.57±1.35 | **20.61±6.78** | – |
| GSE OT | **38.58±0.61** | **1.38±0.65** | **47.32±0.27** | 21.28±7.98 | – |
| $K$-cluster | 3.74±0.23 | **0.48±0.69** | 10.62±0.96 | **29.58±1.45** | – |
| GSE $K$-cluster | **26.92±0.86** | -6.03±0.24 | **30.76±1.27** | 31.50±2.00 | – |

(d) Concepts

| Method | PE | WG-PE | % Feas. | Robustness | WC Robustness |
|---|---|---|---|---|---|
| DiCE | **8.10±0.07** | **3.03±0.08** | 76.85±0.47 | 20.59±3.95 | 26.08 |
| GSE DiCE | 7.65±0.08 | 2.41±0.03 | **77.52±0.47** | **14.32±3.30** | **18.97** |
| OT | **73.83±0.25** | 53.59±0.40 | 82.77±0.16 | 35.69±8.93 | 48.27 |
| GSE OT | 71.58±0.25 | **67.52±0.05** | **100.00±0.00** | **28.30±2.10** | **31.14** |
| $K$-cluster | **28.28±0.48** | 16.31±0.09 | 85.57±0.47 | 81.42±50.15 | 151.45 |
| GSE $K$-cluster | 25.93±0.02 | **20.75±0.06** | **92.62±0.00** | **37.90±27.15** | **91.64** |

distribution is unknown and the goal is to improve model robustness to unseen out-of-distribution data. In contrast, domain adaptation aims to adapt a model learned on the source distributions to some *known target distribution*. But similar techniques were proposed for domain generalization and domain adaptation, including augmenting training data (Li et al., 2021; Yao et al., 2022; Motiian et al., 2017), adding regularization terms to the loss function (Zhao et al., 2020; Balaji et al., 2018; Kim et al., 2021; Cicek & Soatto, 2019; Saito et al.) and meta-learning (Li et al., 2018; Motiian et al., 2017). There are also many real world distribution shift datasets such as the iWildCam dataset (Beery et al., 2021) and the Camelyon17 dataset of (Bandi et al., 2018) as part of the WILDS datasets (Koh et al., 2021).

## J THEORETICAL ANALYSIS

As mentioned in Theorem 1, to study group irregularities from a theoretical perspective, we consider a simple setting where Wasserstein distance has a closed-form solution. We define our source and

Table 10: Civil Comments featurization ablation

(a) Topic Modeling

| Method | PE | WG-PE | % Feas. | Robustness | WC Robustness |
|---|---|---|---|---|---|
| DiCE | -7.09±0.18 | -6.67±0.19 | 90.56±0.0 | **60.54±3.33** | – |
| GSE DiCE | **-6.50±0.07** | **-6.05±0.12** | **90.89±0.0** | 61.64±4.20 | – |
| OT | -10.11±0.31 | -10.84±0.46 | 76.63±0.60 | **2.19±0.25** | **2.85** |
| GSE OT | **-9.85±0.11** | **-9.85±0.11** | **83.23±0.54** | 2.31±0.27 | 3.08 |
| $K$-cluster | -9.29±0.01 | -8.19±0.01 | 82.53±0.0 | 4.58±1.92 | 8.50 |
| GSE $K$-cluster | **-8.78±0.04** | **-7.65±0.03** | **82.80±0.0** | **2.70±0.84** | **5.19** |

(b) Embedding

| Method | PE | WG-PE | % Feas. | Robustness | WC Robustness |
|---|---|---|---|---|---|
| OT | **81.35±0.21** | 19.32±0.13 | 82.95±0.11 | 1.28±0.06 | 1.32 |
| GSE OT | 67.42±1.16 | **65.19±1.22** | **100.00±0.00** | **1.19±0.05** | **1.25** |
| $K$-cluster | **6.18±0.02** | 4.04±0.00 | 82.88±0.16 | **0.56±0.02** | **0.57** |
| GSE $K$-cluster | 5.21±0.71 | **4.96±0.88** | **85.15±0.57** | 0.65±0.06 | 0.74 |

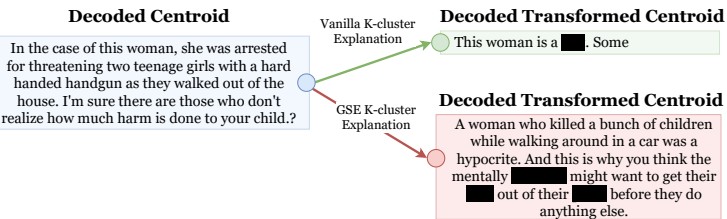

Figure 10: Visualization of a Vanilla and GSE $K$-cluster shift explanation learned for non-toxic to toxic shift in the Civil dataset using an embedding space featurization. Inappropriate words are redacted. The cluster centroid embedding is decoded to text using the embedding inversion model vec2text. This visualized cluster consists mostly of discussions of crimes and legal cases somehow involving women. The vast majority of the comments in this cluster are written by women but get mapped by the Vanilla explanation to a comment that is far from other comments written by women (it contains a slur for a woman). On the other hand, the GSE explanation makes the source centroid more toxic while making it align better with the group of comments written by women.

target distributions as $P = \text{Bernoulli}(p)$ and $Q = \alpha \cdot \text{Bernoulli}(p) + \beta$ where $p \in [0, 1]$ and $\alpha, \beta \in \mathbb{R}$. In this case, our source distribution consists of a $1 - p$ fraction of the data at $x = 0$ and a $p$ fraction of the data at $x = 1$ while the target consists of a $1 - p$ fraction of the data at $x = \beta$ and the other $p$ fraction of the data at $x = \alpha + \beta$. In this one-dimensional case, we will consider a 1-cluster transport explanation introduced by Kulinski & Inouye (2023). Hence, given this context, we can first optimize $\text{PE}(\theta)$ and $\text{WG-PE}(\theta)$ to obtain the value of $\theta_{\text{PE}}$ and $\theta_{\text{WG}}$, i.e.:

**Lemma 2.** *Given the context in Theorem 1, maximizing PE($\theta$) and WG-PE($\theta$) respectively yields the solution $\theta_{PE}$ and $\theta_{WG}$ as follows:*

$$\theta_{PE} = (\alpha - 1)p + \beta, \theta_{WG} = \frac{2\beta(\alpha + \beta - 1)}{\alpha + 2\beta - 1} \tag{10}$$

*Proof.* The overall proof is composed of two steps. The first step explicitly calculates the Wasserstein-2 distance between the source and target distributions, which is followed by the calculation and optimization of $\text{PE}(\theta)$ and $\text{WG-PE}(\theta)$ for Vanilla explanations and GSE explanations respectively.

**Step 1: calculating Wasserstein-2 distance** According to (Kolouri et al., 2019), we can obtain the closed-form solution of the Wasserstein-2 distance between two distributions $P$ and $Q$, i.e.:

$$W_2^2(P, Q) = \int_0^1 \left| F_p^{-1}(u) - F_q^{-1}(u) \right|^2 du, \tag{11}$$

where $F_P$ and $F_Q$ are the Cumulative Distribution Function of the distribution $P$ and $Q$ and thus $F_P^{-1}$ and $F_q^{-1}$ represents the quantile functions of $P$ and $Q$.

Given that $P$ and $Q$ are Bernoulli distributions, $F_P^{-1}$ and $F_q^{-1}$ are derived as follows:

$$F_p^{-1}(q) = \begin{cases} 0 & q < (1-p) \\ 1 & q \geq (1-p) \end{cases} \qquad F_q^{-1}(q) = \begin{cases} \beta & q < (1-p) \\ \alpha + \beta & q \geq (1-p) \end{cases}.$$

By plugging the above formula into equation 11, $W_2^2(P, Q)$ becomes:

$$W_2^2(P, Q) = (1-p)\beta^2 + p(1 - \alpha - \beta)^2 \tag{12}$$

Similarly, for the mapped source distribution $M(P; \theta) = \{x + \theta; x \in P\}$ after we apply the mapping $M$, the quantile function of $M(P; \theta)$ becomes:

$$F_{M(P;\theta)}^{-1}(q) = \begin{cases} \theta & q < (1-p) \\ 1 + \theta & q \geq (1-p) \end{cases}$$

which is then plugged into equation 11 to calculate the Wasserstein-2 distance between $M(P; \theta)$ and $Q$:

$$W_2^2(M(P; \theta), Q) = (1-p)(\theta - \beta)^2 + p(1 + \theta - (\alpha + \beta))^2 \tag{13}$$

By denoting group 1 and 2 from the source distribution $P$ as $P_1$ and $P_2$ and group 1 and 2 from the target distribution $Q$ as $Q_1$ and $Q_2$ respectively, we can also calculate the Wasserstein-2 distance within each group in a similar fashion:

$$W_2^2(P_1, Q_1) = |\beta|^2 \tag{14}$$

$$W_2^2(M(P_1; \theta), Q_1) = |\theta - \beta|^2 \tag{15}$$

$$W_2^2(P_2, Q_2) = |1 - (\alpha + \beta)|^2 \tag{16}$$

$$W_2^2(M(P_2; \theta), Q_2) = |1 + \theta - (\alpha + \beta)|^2 \tag{17}$$

**Step 2: calculating and optimizing PE$(\theta)$ and WG-PE$(\theta)$**    By plugging equation 12 and equation 13 to equation 1, we can further calculate PE$(\theta)$ as follows:

$$\begin{aligned} \text{PE}(\theta) &= 1 - W_2^2(M(P; \theta), Q)/W_2^2(P, Q) \\ &= 1 - \frac{(1-p)(\theta - \beta)^2 + p(1 + \theta - (\alpha + \beta))^2}{(1-p)\beta^2 + p(1 - \alpha - \beta)^2}. \end{aligned}$$

Through some algebraic manipulations, we simplify the above formula as follows:

$$\text{PE}(\theta) = 1 - \frac{(\theta - p(\alpha - 1) - \beta)^2 + p(1-p)(\alpha - 1)^2}{(1-p)\beta^2 + p(1 - \alpha - \beta)^2}.$$

Therefore, the above formula is quadratic with respect to $\theta$ and we maximize PE$(\theta)$ when $\theta_{\text{PE}} := \theta = (\alpha - 1)p + \beta$.

Similarly, to derive the GSE explanations, we can plug equation 14 - equation 17 into equation 5 to obtain the worst-group objective WG-PE$(\theta)$ as follows:

$$\begin{aligned} \text{WG-PE}(\theta) &= \min\left(1 - \frac{W_2^2(M(S_1; \theta), T1)}{W_2^2(S_1, T_1)}, 1 - \frac{W_2^2(M(S_2; \theta), T2)}{W_2^2(S_2, T_2)}\right) \\ &= \min\left(1 - \frac{(\theta - \beta)^2}{\beta^2}, 1 - \frac{((1 + \theta) - (\alpha + \beta))^2}{(1 - (\alpha + \beta))^2}\right) \end{aligned}$$

Note that since both $1 - \frac{(\theta - \beta)^2}{\beta^2}$ and $1 - \frac{((1+\theta) - (\alpha+\beta))^2}{(1-(\alpha+\beta))^2}$ are quadratic to $\theta$, then there are three possible $\theta_{\text{WG}}$ which maximizes WG-PE$(\theta)$, i.e., $\theta = \beta$ which maximizes $1 - \frac{(\theta-\beta)^2}{\beta^2}$, $\theta = \alpha + \beta - 1$ which maximizes the other term while $\theta = \frac{2\beta(\alpha+\beta-1)}{\alpha+2\beta-1}$ which makes these two terms equal[3]. We discuss these three cases respectively as follows.

---

[3]Note that $1 - \frac{(\theta-\beta)^2}{\beta^2} = 1 - \frac{((1+\theta)-(\alpha+\beta))^2}{(1-(\alpha+\beta))^2}$ can also happen when $\theta = 0$. But this means that no shifts happen, which is thus ignored

**Case 1:** $\theta_{\mathbf{WG}} = \beta$ This case happens when $1 - \frac{(\theta-\beta)^2}{\beta^2} \leq 1 - \frac{((1+\theta)-(\alpha+\beta))^2}{(1-(\alpha+\beta))^2}$. By plugging $\theta = \beta$ into this inequality, we can get the following constraints on $\alpha$ and $\beta$:

$$\frac{((1+\theta)-(\alpha+\beta))^2}{(1-(\alpha+\beta))^2} = \frac{(1-\alpha)^2}{(1-(\alpha+\beta))^2} <= 0,$$

which is only valid when $\alpha = 1$, $\theta = \alpha + \beta - 1$ and thus $1 - \frac{(\theta-\beta)^2}{\beta^2} = 1 - \frac{((1+\theta)-(\alpha+\beta))^2}{(1-(\alpha+\beta))^2}$.

**Case 2:** $\theta_{\mathbf{WG}} = \alpha + \beta - 1$ This case happens when $1 - \frac{(\theta-\beta)^2}{\beta^2} \geq 1 - \frac{((1+\theta)-(\alpha+\beta))^2}{(1-(\alpha+\beta))^2}$. By plugging $\theta_{\mathrm{WG}} = \alpha + \beta - 1$ into this inequality, we get the following constraints on $\alpha$ and $\beta$:

$$\frac{(\theta - \beta)^2}{\beta^2} = \frac{(\alpha - 1)^2}{\beta^2} \leq 0,$$

which is only valid when $\alpha = 1$, $\theta = \beta$ and thus $1 - \frac{(\theta-\beta)^2}{\beta^2} = 1 - \frac{((1+\theta)-(\alpha+\beta))^2}{(1-(\alpha+\beta))^2}$.

**Case 3:** $\theta_{\mathbf{WG}} = \frac{2\beta(\alpha+\beta-1)}{\alpha+2\beta-1}$ This case happens when $1 - \frac{(\theta-\beta)^2}{\beta^2} = 1 - \frac{((1+\theta)-(\alpha+\beta))^2}{(1-(\alpha+\beta))^2}$ holds. As the analysis of Case 1 and Case 2 suggests, WG-PE($\theta$) gets maximized when the two terms of WG-PE($\theta$) are equal. Therefore, Case 1 and Case 2 could be regarded as special cases of Case 3. Therefore, this suggests that the Wasserstein-2 distance within each group gets reduced by the same amount when WG-PE is optimized. But it is worth noting that there are two implicit constraints on $\alpha$ and $\beta$, i.e., $\alpha + 2\beta \neq 1$ and $\beta(\alpha + \beta - 1) \neq 0$.

This thus concludes the proof.

$\square$

Given Lemma 2, we can then show the proof of Theorem 1 as follows.

### J.1 PROOF OF THEOREM 1

*Proof.* First of all, by plugging the values of $\theta_{\mathrm{WG}}$ and $\theta_{\mathrm{PE}}$ into PE and WG-PE respectively, we can get the following expressions after some algebraic manipulations:

$$\mathrm{PE}(\theta_{\mathrm{PE}}) = 1 - \frac{W_2^2(M(P;\theta_{\mathrm{PE}}),Q)}{W_2^2(P,Q)} = 1 - \frac{p(1-p)(\alpha-1)^2}{(1-p)\beta^2 + p(\alpha+\beta-1)^2}, \tag{18}$$

$$\mathrm{PE}(\theta_{\mathrm{WG}}) = 1 - \frac{W_2^2(M(P;\theta_{\mathrm{WG}}),Q)}{W_2^2(P,Q)} = 1 - \frac{(\alpha-1)^2}{(\alpha+2\beta-1)^2}, \tag{19}$$

$$\mathrm{WG\text{-}PE}(\theta_{\mathrm{PE}}) = \min\left(1 - \frac{p^2(\alpha-1)^2}{\beta^2}, 1 - \frac{(1-p)^2(\alpha-1)^2}{(\alpha+\beta-1)^2}\right), \tag{20}$$

$$\mathrm{WG\text{-}PE}(\theta_{\mathrm{WG}}) = 1 - \frac{(\alpha-1)^2}{(\alpha+2\beta-1)^2}. \tag{21}$$

The worst-group PE for GSE, i.e., WG-PE($\theta$), can be written without a $\min$. Plus, PE($\theta_{\mathrm{WG}}$) $-$ WG-PE($\theta_{\mathrm{WG}}$) = 0, which thus finished the part of the proof for GSE.

The rest of the proof concerns the comparison between PE($\theta_{\mathrm{PE}}$) and WG-PE($\theta_{\mathrm{PE}}$), we will show that $\theta_{\mathrm{PE}}$ has a discrepancy between the resulting PE and WG-PE in most cases, which we call a *group irregularity*. Since WG-PE($\theta_{\mathrm{PE}}$) involves a minimum of two expressions, $1 - \frac{p^2(\alpha-1)^2}{\beta^2}$ and $1 - \frac{(1-p)^2(\alpha-1)^2}{(\alpha+\beta-1)^2}$, we consider the following three cases: the left expression is less than the right expression, the right expression is less than the left expression, or both expressions are equal.

**Case 1** Since the left expression is less than the right expression, we know that

$$\frac{p^2}{(1-p)^2} > \frac{\beta^2}{(\alpha+\beta-1)^2}.$$

So we can get the following through algebraic manipulation,

$$(\alpha + \beta - 1)^2 > \frac{\beta^2 (1-p)^2}{p^2}.$$

We now lower bound the difference between the overall PE and the worst-group PE:

$$\text{PE}(\theta_{\text{PE}}) - \text{WG-PE}(\theta_{\text{PE}}) = \left( 1 - \frac{p(1-p)(\alpha-1)^2}{(1-p)\beta^2 + p(\alpha+\beta-1)^2} \right) - \left( 1 - \frac{p^2(\alpha-1)^2}{\beta^2} \right)$$

$$= \frac{p^2(\alpha-1)^2}{\beta^2} - \frac{p(1-p)(\alpha-1)^2}{(1-p)\beta^2 + p(\alpha+\beta-1)^2},$$

Then by leveraging the fact that $(\alpha + \beta - 1)^2 > \frac{\beta^2(1-p)^2}{p^2}$, we can derive the lower bound of the above formula:

$$\text{PE}(\theta_{\text{PE}}) - \text{WG-PE}(\theta_{\text{PE}}) > \frac{p^2(\alpha-1)^2}{\beta^2} - \frac{p(1-p)(\alpha-1)^2}{(1-p)\beta^2 + p\left(\frac{\beta^2(1-p)^2}{p^2}\right)}$$

$$= \frac{p^2(\alpha-1)^2}{\beta^2} - \frac{p(1-p)(\alpha-1)^2}{(1-p)\beta^2\left(1+\frac{1-p}{p}\right)} = \frac{p^2(\alpha-1)^2}{\beta^2} - \frac{p^2(\alpha-1)^2}{\beta^2} = 0$$

**Case 2** Since the right expression is less than the left expression, we know that

$$\frac{p^2}{(1-p)^2} < \frac{\beta^2}{(\alpha+\beta-1)^2}$$

so we can get the following through algebraic manipulation,

$$\beta^2 > \frac{(\alpha+\beta-1)^2 p^2}{(1-p)^2}.$$

We now lower bound the difference between the overall PE and the worst-group PE:

$$\text{PE}(\theta_{\text{PE}}) - \text{WG-PE}(\theta_{\text{PE}}) = \left( 1 - \frac{p(1-p)(\alpha-1)^2}{(1-p)\beta^2 + p(\alpha+\beta-1)^2} \right) - \left( 1 - \frac{(1-p)^2(\alpha-1)^2}{(\alpha+\beta-1)^2} \right)$$

$$= \frac{(1-p)^2(\alpha-1)^2}{(\alpha+\beta-1)^2} - \frac{p(1-p)(\alpha-1)^2}{(1-p)\beta^2 + p(\alpha+\beta-1)^2}$$

Then based on the fact that $\beta^2 > \frac{(\alpha+\beta-1)^2 p^2}{(1-p)^2}$, we can derive the lower bound of the above formula as follows:

$$\text{PE}(\theta_{\text{PE}}) - \text{WG-PE}(\theta_{\text{PE}}) > \frac{(1-p)^2(\alpha-1)^2}{(\alpha+\beta-1)^2} - \frac{p(1-p)(\alpha-1)^2}{(1-p)\left(\frac{(\alpha+\beta-1)^2 p^2}{(1-p)^2}\right) + p(\alpha+\beta-1)^2}$$

$$= \frac{(1-p)^2(\alpha-1)^2}{(\alpha+\beta-1)^2} - \frac{p(1-p)(\alpha-1)^2}{p(\alpha+\beta-1)^2\left(\frac{p}{1-p}+1\right)}$$

$$= \frac{(1-p)^2(\alpha-1)^2}{(\alpha+\beta-1)^2} - \frac{(1-p)^2(\alpha-1)^2}{(\alpha+\beta-1)^2} = 0$$

**Case 3** For the final case where the left and right expression are equal, this means that

$$\frac{p^2}{(1-p)^2} = \frac{\beta^2}{(\alpha+\beta-1)^2}.$$

From the analysis in the two above cases, we can see that we have that $\text{PE}(\theta_{\text{PE}}) - \text{WG-PE}(\theta_{\text{PE}}) = 0$, so there is no disparity between the overall and worst-group PE. We can solve the equality

$$\frac{p^2}{(1-p)^2} = \frac{\beta^2}{(\alpha+\beta-1)^2}.$$

for $p$ to determine what values of $p$ end up in this case. Solving for $p$ results in $p = \frac{\beta}{\alpha+2\beta-1}$ or $p = \frac{\beta}{1-\alpha}$.

We have now shown that group irregularities exist for all $p$ for cases 1 and 2 and that group irregularities do not exist for case 3 when $p = \frac{\beta}{\alpha+2\beta-1}$ or $p = \frac{\beta}{1-\alpha}$, which thus concludes the proof. $\qquad\square$

## J.2 ANALYSIS OF ROBUSTNESS AND FEASIBILITY

First we analyze robustness in a 1D setting using a 1-cluster transport explanation and then we analyze feasibility in the same setting but with an optimal transport explanation.

Robustness is formally defined in Section 3.2 as

$$\Omega(\theta; M, P, Q, \epsilon) = \|M(P; \theta) - M(P(\epsilon); \theta(\epsilon))\|_2 / \|P - P(\epsilon)\|_2.$$

We analyze the robustness of an explanation which optimizes PE and one which optimizes WG-PE with the following theorem.

**Theorem 3.** *Given the same setting as that in Theorem 1, the robustness with respect to a small perturbation $\epsilon$ to $p$ is $\Omega(\theta_{PE}; M, P, Q, \epsilon) = O(\alpha)$ while $\Omega(\theta_{WG}; M, P, Q, \epsilon) = 0$.*

*Proof.* We will analyze robustness with respect to a small perturbation $\epsilon$ to $p$, the proportion of samples in each group of the source and target. First, as analyzed in Lemma 2, $\theta_{\text{WG}} = \frac{2\beta(\alpha+\beta-1)}{\alpha+2\beta-1}$ is independent of $p$. Thus, any change to $p$ will not affect the explanation, so $\Omega(\theta_{\text{WG}}; M, P, Q, \epsilon) = 0$. Next, for the regular explanation, $\theta_{\text{PE}} = (\alpha - 1)p + \beta$. Thus,

$$\begin{aligned}
\Omega(\theta; M, P, Q, \epsilon) &= \|(\alpha - 1)p + \beta - (\alpha - 1)(p + \epsilon) - \beta\|_2 / \|\epsilon\|_2 \\
&= \|(\alpha - 1)\epsilon\|_2 / \|\epsilon\|_2 \\
&= \|(\alpha - 1)\|_2 = O(\alpha),
\end{aligned}$$

which concludes the proof. $\qquad\square$

For feasibility, we can perform the analysis in more general settings where the shift explanations are in the form of the optimal transport map. An optimal transport explanation has the form $M(x; \theta) = x + \theta(x)$. Based on the solution to the optimal transport problem in 1D (Rachev & Rüschendorf, 1998), the form of the optimal transport map is:

$$\theta(x) = (F_S)^{-1} \circ F_T(x)$$

where $F_S$ is the cumulative distribution function (cdf) of the source, $F_T$ is the cdf of the target. Thus $(F_S)^{-1}$ is the quantile function of the source distribution. When the source and target are defined as Bernoulli distributions, we have the next lemma.

**Lemma 4.** *Let $\alpha, \beta \in \mathbb{R}$, $p \in [0, 1]$, and define $P = \alpha \cdot Bernoulli(p) + \beta$. Then the cdf and quantile functions for $P$ are the following:*

$$F_P(x) = \begin{cases} 1 - p & \text{if } x = \beta \\ 1 & \text{if } x = \alpha + \beta \end{cases} \qquad F_P(x)^{-1} = \begin{cases} \beta & \text{if } x \leq (1 - p) \\ \alpha + \beta & \text{if } x > (1 - p) \end{cases}.$$

To analyze feasibility, we use the definition of feasibility from equation 2:

$$\% \text{ Feasible} = \left[ \sum_{x \in P} a(x, M(x; \theta)) \right] / \|P\|$$

Let $p \in [0, 1]$, $\alpha, \beta \in \mathbb{R}$ and define $P = \text{Bernoulli}(p)$ and $Q = \alpha \cdot \text{Bernoulli}(p) + \beta$. Define groups for $x \sim P$ as group 1 if $x = 0$ and group 2 if $x = 1$. Define groups for $x \sim Q$ as group 2 if $x = \alpha + \beta$ and group 1 if $x = \beta$.

We can write the optimal transport explanation to explain the shift between $S$ and $T$ using Lemma 4 as:

$$M(x; \theta_{\text{PE}}) = \begin{cases} x + \beta & \text{if } x = 0 \\ x + \alpha + \beta - 1 & \text{if } x = 1. \end{cases}$$

We then find the worst-group optimal transport solution. The worst-group optimal transport is an optimal transport in the worst case over the groups of the source and target data. Thus, we have that this worst-case optimal transport has the following form based on the definition of an optimal transport map (Rachev & Rüschendorf, 1998):

$$\pi^* = \inf_{\pi} \max \left( \int_{x=0, y=\alpha+\beta} \|x - y\|_2^2 d\pi(x, y), \int_{x=1, y=\beta} \|x - y\|_2^2 d\pi(x, y) \right)$$

where $\pi$ is a valid transport map. Since the subgroups are point distributions, the optimal transport between the two point groups has a direct solution which is to map the source point to the target point. Thus, the worst-group optimal transport is optimal for both groups as seen by the optimal transport explanation shown below:

$$M(x; \theta_{\text{WG}}) = \pi^* = \begin{cases} x + \alpha + \beta & \text{if } x = 0 \\ x + \beta - 1 & \text{if } x = 1. \end{cases}$$

**Theorem 5.** *Let $p \in [0, 1]$, $\alpha, \beta \in \mathbb{R}$ and define $P = Bernoulli(p)$ and $Q = \alpha \cdot Bernoulli(p) + \beta$. Define groups for $x \sim P$ as group 1 if $x = 0$ and group 2 if $x = 1$. Define groups for $x \sim Q$ as group 2 if $x = \alpha + \beta$ and group 1 if $x = \beta$. Let $M(x; \theta_{PE})$ be an optimal transport explanation and $M(x; \theta_{WG})$ be a worst-group optimal transport explanation. Then, $M(x; \theta_{PE})$ has feasibility of 0 while $M(x; \theta_{WG})$ has feasibility of 1.*

*Proof.* In the definition of Feasible, we define

$$a(x, y) = \begin{cases} 1 & \text{if } x = 0 \text{ and } y = \alpha + \beta \text{ or } x = 1 \text{ and } y = \beta \\ 0 & \text{otherwise.} \end{cases}$$

This definition encapsulates the desire to keep samples from group 1 in the source in group 1 in the target and the same for group 2. We can immediately see that $X_{\text{PE}}$ maps all samples from group 1 in the source to group 2 in the target, so feasibility will be 0. On the other hand, $X_{\text{WG}}$ maps samples respecting the defined group structure, so feasibility is 1. $\square$

## K    LIMITATIONS AND SOCIETAL IMPACTS

GSE explanations are only as good as the underlying shift explanation method. For instance, $K$-cluster transport can result in weak explanations that minimally reduce the Wasserstein distance between the source and target distributions if too few clusters are used (i.e. $K$ is chosen too small). On the other hand, the Optimal Transport explanation that we found reduced the Wasserstein distance the most, is not very interpretable since each source sample can be mapped differently. This results in the explanation being interpretable only on a per-sample basis. Improved interpretability of shift explanations is an area for future work.

In addition to interpretability of the explanation, our shift explanations for image and language data rely on interpretable feature extraction methods and methods for counterfactual modification based on changes to the features as described in Section 3.4.2. We designed a system for interpretable feature extraction which uses a bag-of-words feature representation, but this method looses the context that words are used in and it is difficult to make counterfactual modifications. Creating disentangled embedding spaces for interpretable embeddings that can also be used for counterfactual modification is an area of active research, but there is still work left to make these approaches more general.

We also found that GSE is sensitive to the choice of groups. Even though unsupervised methods can be used to select groups as shown in Appendix E, future work can look at how to best select or design groups. For instance, it may be the case that we know of some groups, but we want the rest of the data to be grouped appropriately.

Finally, while we evaluated the worst-case robustness, our method sometimes results in worse worst-case robustness than the vanilla approach. This is again due to the choice of groups. Future work should investigate how to extend group robustness to worst-case group robustness of shift explanations so that a bad choice of groups does not negatively impact robustness.

Explanations which look plausible but are actually wrong can be harmful. This creates the illusion of understanding, and this can have serious downstream implications especially if policies are constructed from a shift explanation. With this work we hope to uncover some properties that a good shift explanation should have and design metrics and learning procedures based on group robustness to rectify these issues.

