# OpenReview forum: "Rectifying Group Irregularities in Explanations for Distribution Shift"
_ICLR.cc/2024/Conference — Submitted to ICLR 2024_

### Official Review · Reviewer_AAG5 · 2023-10-28

**Soundness:** 3 good
**Presentation:** 2 fair
**Contribution:** 2 fair
**Rating:** 5
**Confidence:** 2

**Summary:**

This paper proposes Group-aware Shift Explanations (GSE) to address the group irregularities in explanations for distribution shift. The key idea is to optimize for the worst-group PercentExplained (PE). Experimental results justify the effectiveness of the proposed method.

**Strengths:**

- The problem of group irregularities proposed is interesting.

**Weaknesses:**

- The proposed method seems quite limited as it works under the assumption that the source and target are already partitioned into disjoint groups and the correspondence of the groups is available. While for many problems in practice (e.g., domain adaptation for semantic segmentation), such information is not available, which is a major challenge for many problems in CV and NLP.
- The extension to image/language data is naive and is more similar to how to use existing methods or pre-trained models to convert such data into tabular data.

**Questions:**

Please see weaknesses above.

---

> ### Author Response · Authors · 2023-11-18
>
> Thank you for reading our paper and for the review. Our response is below:
>
> ## Requirement of group annotations
> In Section 4.1 of our initial submission, we briefly mention that our method can apply to scenarios without group labels and we included an experiment in Appendix E. We find that using KMeans to determine groups results in improving the feasibility and robustness of explanations similar to when we use prespecified groups.
>
> ## Extension to image/language data as naive
> Regarding the point that the extension to image/language data was naive, we want to emphasize that the way of transforming image and language data into interpretable features in our method is just one way of generating human-understandable shift explanations. In the updated paper, we further expanded our framework by considering two additional methods of generating shift explanations for image and language data. These two methods are the following:
>
> 1. Deriving shift explanations in a pre-trained model’s embedding space and decoding the embedding into human-understandable explanations using a pre-trained vec2text [Morris et al, 2023] model.
> 2. Using concepts (see [Koh et al, ICML 2020]) as the interpretable features for generating shift explanations for image data.
>
> We performed additional experiments on language and image data with these two methods and included the results in Appendix H. The results again demonstrate the benefits of GSE in rectifying group irregularities and enhancing feasibility and robustness in generating shift explanations. In addition, to the best of our knowledge, other advanced techniques for generating shift explanations for image and language data do not yet exist. The only other work on shift explanations is from Kulinski et al. [Kulinski et al, ICML 2023] which also considered shift explanations over bag-of-word features for language.
>
> Finally, we emphasize that instead of proposing new methods for extracting interpretable features from different data modalities, we wanted to show that group irregularities exist when using shift explanations over different modalities and that our method is able to alleviate this issue. We hope that further research can use our framework to propose more sophisticated shift explanation methods for images and data.
>
> [Morris et al, 2023]: Morris, John X., et al. "Text embeddings reveal (almost) as much as text." arXiv preprint arXiv:2310.06816 (2023).
>
> [Kulinski et al, ICML 2023]: Sean Kulinski and David I Inouye. Towards explaining distribution shifts. In International Conference on Machine Learning, pp. 17931–17952. PMLR, 2023.
>
> [Koh et al, ICML 2020] Koh, Pang Wei, Thao Nguyen, Yew Siang Tang, Stephen Mussmann, Emma Pierson, Been Kim, and Percy Liang. "Concept bottleneck models." In International conference on machine learning, pp. 5338-5348. PMLR, 2020.

---

> > ### Author Response · Authors · 2023-11-22
> >
> > We believe we have addressed all your concerns, but if there is anything remaining, we are happy to respond in the remaining discussion period. We would also appreciate any consideration of adjusting your score in light of the improvements and clarifications made in our response. Thank you!

---

### Official Review · Reviewer_LTxx · 2023-10-29

**Soundness:** 2 fair
**Presentation:** 3 good
**Contribution:** 2 fair
**Rating:** 5
**Confidence:** 4

**Summary:**

For the explanation of distribution shift, e.g., between train and test datasets, existing methods often find optimal transport between two datasets to quantify the shift. This paper considers group-based optimization and proposes to consider the distance for the worst group to achieve a reasonable explanation (i.e., avoiding overall optimal but locally undesirable transport). Experiments with tabular, language, and image datasets show reasonable explanations of the proposed method using several metrics, including the feasibility metric.

**Strengths:**

+ The explainability of domain shift is a core problem in machine learning that could be more actively studied.
+ The idea of introducing group robustness to the explanation of domain shift is quite natural and reasonable.
+ The experiment shows broad categories of datasets in which the proposed method is ready to use in practice.

**Weaknesses:**

- Considering worst-case loss is quite a natural idea and is used in many contexts, as the paper also mentions in the related work section. This kind of loss can be easily plugged into the Wasserstein distance minimization. Readers may consider the proposed method a pure application of these methods to a specific method (i.e., the distribution shift explanation method by Kulinsky & Inouye, 2023).

- I consider that penalizing the loss of the worst group would be a simple way of achieving distributionally robust optimization (DRO), as representative related papers using worst-case losses call their methods distributionally robust (e.g., Sagawa et al., 2019). In such a sense, it is unnatural that this paper does not discuss anything about DRO. Readers would want to see the discussions about how their method can be stated as a new method of DRO for optimal transport problems.

- (Cont'd) In such a sense, we can easily find extensive studies regarding Wasserstein DRO (WDRO) such as:

    - Kuhn, Daniel, et al. "Wasserstein distributionally robust optimization: Theory and applications in machine learning." Operations research & management science in the age of analytics. Informs, 2019. 130-166.
    - Kwon, Yongchan, et al. "Principled learning method for Wasserstein distributionally robust optimization with local perturbations." International Conference on Machine Learning, 2020.

**Questions:**

How is the proposed method related to Wasserstein distributionally robust optimization methods?

---

> ### Author Response · Authors · 2023-11-18
>
> Thank you for reading our paper and for the helpful feedback. We respond to your comments and questions below:
>
> ## Discussion of DRO
> Thank you for the suggestion to discuss DRO, we added a short discussion to Section 5. Our formulation of the worst-group loss is a form of DRO for learning distribution shift explanations. While DRO is often used for reducing spurious correlations in a discriminative model, we use it for finding shift explanations that can alleviate the group irregularity issue and enhance feasibility and robustness at the same time. In our response to reviewer XuMW, we demonstrate that improving the feasibility of the shift explanations can be tied to mitigating spurious correlations. Furthermore, leveraging worst-group loss can not only contribute to reducing spurious correlations but also yield the additional benefit of enhancing the robustness of the  generated shift explanations. To substantiate these claims, we further provide theoretical analysis (Section 3.5 in the revision) and comprehensive empirical evaluations in the paper.
>
> ## Connection to Wasserstein DRO
> Regarding the connection between our approach and Wasserstein DRO, despite the necessity of optimizing Wasserstein distance in both our setting and prior studies such as [Kuhn, Daniel, et al], we address an orthogonal problem from them. In those prior studies, Wasserstein DRO is a replacement for ERM which optimizes a risk over all distributions within a small Wasserstein ball for *learning a more robust model*. On the other hand, as pointed out in [Kulinski et al, ICML 2023], the goal of optimizing Wasserstein distance in our setting is for *generating explanations for understanding how distribution shift occurs*.  Plus, different from [Kulinski et al, ICML 2023], our primary focus is on investigating the group irregularity issues in shift explanations learned by optimizing reduction in Wasserstein distance. We added some more discussion concerning this point in Section 5 in the updated paper.
>
> [Kulinski et al, ICML 2023]: Sean Kulinski and David I Inouye. Towards explaining distribution shifts. In International Conference on Machine Learning, pp. 17931–17952. PMLR, 2023.

---

> > ### Author Response · Authors · 2023-11-22
> >
> > We believe we have addressed all your concerns, but if there is anything remaining, we are happy to respond in the remaining discussion period. We would also appreciate any consideration of adjusting your score in light of the improvements and clarifications made in our response. Thank you!

---

### Official Review · Reviewer_3qo2 · 2023-10-30

**Soundness:** 3 good
**Presentation:** 3 good
**Contribution:** 2 fair
**Rating:** 5
**Confidence:** 2

**Summary:**

The manuscript addresses the problem of explainable distribution shifts. For distributions covering changeable and unchangeable properties,  computing the shift blindly will lead to infeasible solutions. This is avoided by grouping the samples, i.e, maintaining the unchangeable properties as constraints. The proposed method is evaluated on several datasets, most of which containing language data or based on language data.

**Strengths:**

S1. The manuscript is well structured and illustrated.

S2. The overall problem of infeasible distribution shifts is explained exhaustively.

S3. An attempt is made to open up the method towards other types of data, e.g. images.

**Weaknesses:**

W1. Strong statements in the manuscript frequently lack references. Examples: beginning of abstract (if references are to be avoided here, the wording should be repeated in the introduction together with proper references), first mentioning of "shift explanation", example in figure 1, percentages given on page 3 (bottom).

W2. In many places, sentences are formed that do not make sense or are not sufficiently accurate. Examples: "map a male close to a female by changing the age feature", delineations between contributions 2 to 4, caption figure 2, explanation of setting for figure 3b.

W3. Theorem 1 (and its proof) mixes general parameters alpha and beta and numbers (10) and the meaning of the parameters / 10 is not well-defined in the theorem.

W4. The "generalization to image data" remains effectively a pure language problem. CLIP and stable diffusion are used as "translators" from image to text and back, at the text level no image-specific properties remain.

**Questions:**

Q1. Why is it not possible to model the problem using conditional distributions and shifts between those?

Q2. Why is sex considered unchangeable whereas age is changeable?

Q3. Why using alpha = beta =10 in theorem 1?

Q4. "Robustness" is defined in terms of changes of feasibility. Why is this aspect not covered by statistical variations over multiple runs?

The authors provided a good rebuttal and addressed most questions well, leading to an updated assessment.

**Details Of Ethics Concerns:**

Not sure how serious this is: In the text examples for group membership, sex is considered to be a permanent feature and a change of sex as an infeasible solution. This is probably intended as a pedagogical example only, but the authors could have chosen a better one, avoiding the risk of some readers feeling discriminated.

---

> ### Comment · Area_Chair_ZZNJ · 2023-11-14
> **Ethics concerns**
>
> Dear authors,
>
> Reviewer 3qo2 raised concerns about the text examples for group membership. Would you be able to use another example to address the concerns?
>
> Thanks,
> Your AC

---

> ### Author Response · Authors · 2023-11-18
>
> Thank you for the review! We respond to your points below:
>
> ## Modeling the problem using shifts between conditional distributions
> We model the problem as explaining shifts between distributions which also respect subgroups. A subgroup is a conditional distribution; for instance, a group can be the source distribution conditioned on age being over 30. We do not use the term conditional distribution since the term “group” is more common in fairness and robustness literature [Dwork et al, ITCS 2012; Sagawa et al, ICLR 2020].
>
> ## Statements lacking references
> We believe that everything mentioned was in fact cited. Everything mentioned in the abstract is cited in the beginning of the introduction. We are not aware of any ICLR papers with citations in the abstract. We also cite the Adult dataset in the introduction (last paragraph of page 1) before it is mentioned in the caption of Figure 1. The percentages at the end of page 3 explicitly refer to the explanation shown in Figure 3a which is cited in the same sentence with the percentage. If there are other places where you believe we are lacking references, please let us know and we will be happy to update the paper.
>
> ## Confusing and imprecise sentences
> The sentences mentioned are all well formed and accurate for their context. We updated the paper to further clarify these sentences or removed them if they were not necessary. The “map a male close to a female…” example appears in the introduction and explains a phenomenon at a high level, but it is still an accurate description. Since this caused confusion, we decided to remove it. For contributions 2-4, we clarified their differences in the paper, we explained “quality” in the caption of Figure 2, and we added a sentence to clarify the setting of Figure 3b.
>
> ## Theorem
> We removed all numbers and explained the parameters in Theorem 1 based on your suggestion. By removing the numbers, we generalized the Theorem. Also, as suggested by reviewer XuMW, we further provided additional theorems to prove that our method can enhance the feasibility and robustness of shift explanations.
>
> ## Generalization to image data
> Regarding the generalization to image data remaining a pure language problem, we only introduce such a translation of image data to language data to produce interpretable explanations, but our method does not depend on this featurization technique. In addition, even though we are using language features, this does not mean that image properties are not represented. Language can capture many image-specific properties such as brightness, perspective, and style which are also used by the text-to-image model. For instance, in Figure 5 of the original submission, the generated caption includes “zoo photography” which is an image style feature
>
> Appendix G of the initial submission also includes experiments where we learn an explanation directly over image pixels to show that our method is not dependent on this featurization step, but the resulting explanations are not interpretable. We have modified Section 3.4.2 to reference these additional experiments in Appendix H. We hope that future work proposes more sophisticated interpretable image featurization techniques so we can use them within our framework to get better image shift explanations.
>
> ## Choice of feasible and infeasible features
> Regarding why we considered the sex feature unfeasible and age feasible, we made this choice based on existing work on feasibility of explanations [Poyiadzi et al, AAAI 20] where sex was treated as immutable and age was partially mutable. Therefore, we allowed age to be modified and deemed the sex feature as infeasible.
>
> We also want to address the ethics issue. It was not our intention to make anyone feel targetted or discriminated against and we apologize if that happened. The choice of sex as infeasible comes from prior work on feasible explanations [Poyiadzi et al, AAAI 2020; Mothilal et al, FAT* 2020]. We do not consider sex as generally infeasible, but we acknowledge that there are cases where a user may not want an explanation to include the sex feature. For instance, [Salimi et al, ECAI 2023] conducted a user study on existing explanation techniques and a user commented that “it is highly unethical to suggest someone changes their sex.” The choice of infeasible features is entirely dependent on a user’s intentions, and we chose to use sex as one example. Instead of sex as an infeasible feature, we can use race or age. In our revision, we currently replaced the example with one using race as an infeasible feature, but we also included an example with age as infeasible in Appendix F, and if the reviewer prefers age then we can swap the examples.

---

> > ### Author Response · Authors · 2023-11-18
> >
> > ## Robustness compared to statistical variations between runs
> > Robustness and statistical variations over multiple runs are two very different aspects of the explanation. We agree that explanations have some small amount of variability over multiple runs of optimization (which is reflected in the error bars in Table 1), but this is independent of the variability of the explanation after small perturbations to the source distribution. In addition, robustness is not defined in terms of changes to feasibility, but we do give an example in Section 2.1 where the lack of robustness leads to changes in an explanation which makes a feasible explanation infeasible. In addition, we added a theoretical analysis of robustness in the 1D setting of Theorem 1 to Appendix J which shows that with no statistical variation between runs (since the optimization has a closed-form solution) there is still a difference in robustness between the regular and our GSE method.
> >
> > [Dwork et al, ITCS 2012]: Dwork, Cynthia, et al. "Fairness through awareness." Proceedings of the 3rd innovations in theoretical computer science conference. 2012.
> >
> > [Sagawa et al, ICLR 2020]: Sagawa, S., Koh, P. W., Hashimoto, T. B., and Liang, P. Distributionally robust neural networks for group shifts: On the importance of regularization for worst-case generalization. In International Conference on Learning Representations (ICLR), 2020.
> >
> > [Poyiadzi et al, AAAI 20]: Rafael Poyiadzi, Kacper Sokol, Raul Santos-Rodriguez, Tijl De Bie, and Peter Flach. Face: feasible and actionable counterfactual explanations. In Proceedings of the AAAI/ACM Conference on AI, Ethics, and Society, pp. 344–350, 2020.
> >
> > [Mothilal et al, FAT* 2020]: Mothilal, Ramaravind K., Amit Sharma, and Chenhao Tan. "Explaining machine learning classifiers through diverse counterfactual explanations." Proceedings of the 2020 conference on fairness, accountability, and transparency. 2020.
> >
> > [Salimi et al, ECAI 2023]: Salimi, Pedram, et al. "Towards Feasible Counterfactual Explanations: A Taxonomy Guided Template-Based NLG Method." ECAI 2023. IOS Press, 2023. 2057-2064.

---

> > > ### Comment · Reviewer_3qo2 · 2023-11-19
> > >
> > > While I do agree that robustness and _stability_ (e.g. to noise / small perturbations) are different things, this does not mean that only stability can be evaluated as statistical variations. If a violation of an underlying assumption leads to an outlier of the prediction result, the model shows a lack of robustness. Outliers lead to statistical variations.

---

> > > > ### Author Response · Authors · 2023-11-20
> > > >
> > > > Thank you for the response. We have posted a further revision of the paper to address your comments and provide a detailed response below.
> > > >
> > > > ## Conditional density and group terminology
> > > > In the original submission and our revisions we included that “group” was interchangeable with “subpopulation” which we believed was appropriate for a broad audience. Based on your suggestion, we added a note in the last paragraph of the introduction that “group” is also interchangeable with “conditional density.”
> > > >
> > > > ## References
> > > > The second sentence of the introduction in our original and revised submissions restates the claim of the first sentence of the abstract with a citation. Sentences three and four of the introduction then provide further references. After carefully reviewing the introduction, we have added four additional citations, including a citation for the first sentence of the introduction (IID assumption), a citation for an early paper mentioning the importance of understanding distribution shift, and a citation for our first mention of “shift explanation”. For the percentages on page three of the original submission, we see that it was not explicitly stated that Figure 3 referred to explanations that we learned in our own experiments. These percentages were removed from the subsequent revision, but based on this feedback, we have added clarifications to the caption of Figure 1 and Section 2 that Figures 1 and 3 refer to our own results which we have also included in Appendix F.1.
> > > >
> > > > ## Contributions 2 and 3
> > > > To clarify, contribution 2 refers to our method, GSE, for learning shift explanations using group information. Contribution 3 refers to the framework of a mapping function (such as a transport map or counterfactual map) and a loss function (such as PE) as the components for learning a shift explanation. This framework not only encompasses existing shift explanation techniques, but allows us to use existing counterfactual explanation techniques as shift explanations which, to our knowledge, has not been previously done.
> > > >
> > > > ## Image-based generalization
> > > > Based on your suggestion, we have modified the image data generalization paragraph in Section 3.4.2 to further clarify that the pipeline in Figure 4 is just one instantiation of the featurization and reverse featurization steps to get interpretable explanations for images. By “Generalizing to image data” we do not intend to solve the current problems with image featurization, but just to show how we can learn explanations over images if we have a featurization method. We discussed the limitations of image featurization in the Limitations section of our original submission. In addition to extracting interpretable features from images, we also performed experiments by using raw pixels as features, which were included in Table 8(a) in Appendix H in the original submission. But based on your suggestion we added a short comparison of the pixel feature experiment with the text feature experiment in Appendix H and added a paragraph to Section 4.3 referencing this experiment and comparison.
> > > >
> > > > ## Robustness and statistical variations
> > > > We want to clarify terminology since robustness and stability are the same in the context of evaluating *explanations*. For explanations, both refer to the extent to which similar inputs result in similar explanations, which we mention in Section 2.1 of the original and revised submission and also formally defined in Section 3.4 of the original and Section 3.2 of the revised submission. This definition is from [Alvarez-Melis & Jaakkola, 2018]. In regards to “outliers of the prediction result,” we believe this may correspond to our metric of worst-case robustness which we reported results for in Table 2 where we evaluate the robustness of shift explanations by adding adversarial perturbations to the source distributions. Variability from the optimization procedure for learning an explanation is a separate issue, but we report error bars over three trials to quantify its effect. Finally, we prove in Theorem 3 in Appendix J that an explanation technique can lack robustness even with no variability from optimization since the optimal explanation has a closed form solution.
> > > >
> > > > [Alvarez-Melis & Jaakkola, 2018]: David Alvarez-Melis and Tommi S Jaakkola. On the robustness of interpretability methods. arXiv preprint arXiv:1806.08049, 2018.

---

> > > > > ### Comment · Reviewer_3qo2 · 2023-11-20
> > > > >
> > > > > Several of my concerns are now addressed. Regarding the last point, I understand (again) the use of terminology in a certain part of the community, but it is important to remember the wider audience of readers.

---

> > > > > > ### Author Response · Authors · 2023-11-20
> > > > > >
> > > > > > We are glad we could address some of your concerns. For the robustness terminology, we defined robustness at a high level (Section 2.1 of the original and revised submission) and provided a formal definition (Section 3.4 of the original and Section 3.2 of the revised submission) with citations to where it was first defined in prior work. To remove any confusion around this terminology, we have updated Section 3.2 to clarify that only the measure of worst-case robustness is comparable to typical notions of *model* robustness which often refers to the extent to which model output changes in face of small perturbations to the input in the worst case, or when the perturbation is assumption violating.

---

> > > > > > > ### Author Response · Authors · 2023-11-22
> > > > > > >
> > > > > > > We believe we have addressed all your concerns, but if there is anything remaining, we are happy to respond in the remaining discussion period. We would also appreciate any consideration of adjusting your score in light of the improvements and clarifications made in our response. Thank you!

---

> ### Comment · Reviewer_3qo2 · 2023-11-19
>
> Conditional distributions: the potential readers of the manuscript will have background from other areas as well and explaining once that "group" and "conditional density" can be used interchangeably will improve the readability significantly.
>
> References: It is the authors' responsibility to be accurate with citations. To this end, I commented examples where I found that citations were too inaccurately used. It is not sufficient to have a reference somewhere that could be the same statement as the first sentence in the abstract (note that I did not insist on adding a reference there but rather to repeat the sentence in the introduction with citation). Regarding the percentages (p. 3), it is not relevant that they are in the figure - it must be said where these numbers are from (= reference or own experiment).
>
> I appreciate the improved formulations in various places. However, I still don't get the understanding between contributions 2 and 3. Contribution 4 is just about the necessary experiments to confirm contribution 2.
>
> The new formulation of the theorem is more relevant.
>
> Image-based generalization: Where does the proposed pipeline include image features ("not interpretable")? The hard part of image representation is the extraction of a descriptive text; however, there are also features that cannot be described by words. A proper image-based generalization should also be in the position to handle those non-textual features. To this end: consider including a brief version of the "raw pixel" experiment, including a short description and a comparison to the text-based version in the main paper.
>
> Regarding the feasible features: maybe a clarification had been sufficient to avoid a potential issue here; also not sure that the wording "race" is more appropriate in that respect.

---

### Official Review · Reviewer_XuMW · 2023-10-31

**Soundness:** 2 fair
**Presentation:** 2 fair
**Contribution:** 3 good
**Rating:** 6
**Confidence:** 3

**Summary:**

This work proposes Group-aware Shift Explanations (GSE), a shift explanation method for understanding distribution shifts. The authors first identify group irregularities as a class of problems in existing shift explanation literature, and then introduce GSE, which utilizes worst-group optimization to rectify such group irregularities. A unified framework is also developed to generalize GSE from K-cluster transport to broad types of existing shift explanation methods, and from tabular data to language and image data. Experiments on tabular, language, and image datasets demonstrate that GSE preserves group structures and mitigates the feasibility and robustness of the state-of-the-art shift explanation approaches.

**Strengths:**

1. This work is the first shift explanation method that identifies group irregularities as a problem that negatively affects the distribution shift explanation ability of the existing approaches, both theoretically and empirically.

2. To rectify the group irregularity issues in existing shift explanation approaches, the authors propose GSE, which leverages the worst-group optimization to optimize the worst-group PercentExplained (PE). GSE maintains group (subpopulation) structures and generates more feasible and robust shift explanations.

3. The authors did a great job adapting the counterfactual explanation methods to the shift explanation setting, developing a general framework that applies GSE to optimal transport and counterfactual explanation methods for generating more reliable shift explanations.

4. Extensive experiments on real-world tabular, language, and image datasets demonstrate the superior performance of GSE in producing more feasible and robust shift explanations while preserving group structures, both quantitatively and qualitatively.

**Weaknesses:**

1. The background introduction should be more clear, especially the parts related to feasibility and robustness. The authors mention several times "feasibility'' and "robustness'' when introducing the significance of group irregularity issues in shift explanation and motivation, but the formal definitions of these two terms are not introduced until Section 3.4, which may lead to confusion about these two terms. I think it would be better if the authors discuss in detail about "feasibility'' and "robustness'' in the introduction and motivation parts. Further, Figure 1(a) (Figure 3(a)) is not clear in illustrating the group irregularities and GSE's solution compared to Figure 1(b) (Figure 3(b)).

2. GSE rectifies the group irregularity issues by simply extending the optimization of PE to optimizing the worst-group PE. Such idea for tackling subpopulation shifts has already been proposed in [1], though it is not designed for shift explanations. Therefore, the idea of GSE is not very novel. Further, the authors only provide theoretical analysis in a simple 1D setting to illustrate the existence of group irregularities. Regarding the proposed GSE, there is no theoretical justification for why GSE generates a more feasible and robust shift explanation while maintaining group structures than the existing methods. It would be better if the authors could provide further theoretical analysis.

3. GSE assumes that the group information is known in the training data, which is hard to satisfy for most real-world scenarios.

4. The authors did a great job of introducing the works related to explaining distribution shift and worst-group robustness. However, for the works related to domain generalization and adaptation, it would be great if they could discuss connections between domain generalization and adaptation and shift explanations.

[1] Sagawa, S., Koh, P. W., Hashimoto, T. B., \& Liang, P. (2019). Distributionally robust neural networks for group shifts: On the importance of regularization for worst-case generalization. arXiv preprint arXiv:1911.08731.

**Questions:**

1. Please see the questions mentioned in Weaknesses.

2. As GSE requires pre-specified groups in the training data to perform worst-case optimization, I wonder how to select the proper feature that divides the group. In the paper, it seems that using unactionable features (e.g., sex) might be an option. I wonder if the authors could discuss further how to divide data into groups.

3. It is known that group imbalance (irregularity) will make the empirical risk minimization models learn spurious correlation, which is vulnerable under distribution shifts. Therefore, leveraging worst-case optimization can mitigate the spurious correlation issue. As GSE also utilizes worst-case optimization to rectify the group irregularities, I wonder if spurious correlations also cause infeasible or vulnerable shift explanations. It would be great if the authors could discuss further the connection between spurious correlations and feasible and robust shift explanations.

---

> ### Author Response · Authors · 2023-11-18
>
> Thank you for taking the time to read and provide feedback on our paper! Our response is below:
>
> ## Regarding the novelty of the proposed method
> We appreciate the pointer to prior work [1] which tackles subpopulation shifts with worst-group loss. The scope of our paper, however, is not to solve the classical subpopulation shift issue as in [1] where model performance on subpopulations is the primary focus. Instead, our goal is to reveal that poor performance on subgroups occurs in distribution shift explanations and propose to leverage worst-group loss to alleviate this issue. In our paper, instead of just extending the optimization of overall PE to the worst-group PE, we further provide theoretical analysis and extensive empirical evidence to demonstrate that optimizing worst-group PE can enhance the feasibility and robustness of distribution shift explanations. To our knowledge, this has not been explored by any prior works. We have expanded on the comparison with DRO in Section 5 and highlighted our theoretical analysis in Section 3.5 in the updated paper.
>
> ## Use of “feasibility” and “robustness” in the introduction and motivation
> Thank you for the feedback on clarifying the discussion of feasibility and robustness. We added additional details to the high-level description of feasibility in Section 2.1 based on your feedback. Intuitively, feasibility measures the percentage of source samples that are modified in an actionable way by the explanation. For instance, if the explanation modifies the race feature (which we may consider as unactionable) for half of the source samples, then the feasibility is 0.5. Robustness measures the degree to which the shift explanation changes with respect to small perturbations to the source distribution. If there are other confusions caused by our high level definitions in Section 2.1, we will be happy to modify the text to make it clearer.
>
> We also admit that the formal definitions of “feasibility” and “robustness” given in Section 3.4 in the original submission came too late. To address this issue, we move Section 3.4 right after Section 3.1 so that the reader can immediately get familiar with these definitions after the motivating examples.
>
> ## Clarity of Figure 1a and 3a
> Based on the feedback from reviewer 3qo2, we changed the example in Figure 1 to no longer use the sex feature. Figure 1a is meant to illustrate how the Vanilla explanation can change the proportions of Black and White people, shown as black and white figures respectively, while the GSE explanation is depicted as keeping the proportion the same. Figure 1a is a comparison of Vanilla and GSE explanations on the global level and we have added a subcaption to clarify this. The other part of the comparison happens with the annotated “Reduction in Wasserstein distance” in Figure 1a where the GSE explanation is shown to have a total reduction in Wasserstein distance very similar to the reduction for the Black subpopulation while the Vanilla explanation has a larger gap between these two reductions. We explain this at the bottom of page 1.
>
> ## Theoretical justification of GSE improving feasibility and robustness
> We added two additional theorems to analyze robustness and feasibility in a 1D setting in Appendix J. Theorem 2 shows that robustness of a regular explanation scales with a parameter of the target distribution, which is worse than the robustness of a worst-group explanation which is 0. Theorem 3 shows that the feasibility of a regular explanation can be 0 while the feasibility of a worst-group explanation is 1. We have limited the theoretical analysis to the 1D setting since we wanted to validate that group irregularities occur even in the simplest 1D case and that worst-group optimization can mitigate the problem. More comprehensive theoretical analysis beyond 1D settings is left for future work.
>
> ## Connection to domain generalization and adaptation
> We added a short comparison of domain generalization and adaptation with shift explanations to the related work. Both domain generalization and adaptation are methods for learning models while shift explanations are concerned with explaining how data shifts when distribution shifts happen.

---

> > ### Author Response · Authors · 2023-11-18
> >
> > ## Choice of groups
> > We want to clarify that our method does not necessarily require annotated group labels and in Appendix E (mentioned in Section 4.1) of the initial submission, we have discussed an experiment without group labels where GSE still improves feasibility and robustness. This method uses a clustering technique to define groups based on feature space similarity. For manually dividing data into groups, we give examples in Section 4.1 of using unactionable features, existing subpopulations such as animal breeds or geographic regions, and domain knowledge of constraints such as properties of cells. We also discuss the selection of groups in the Limitations section (Appendix K) where we acknowledge that groups may not always be available, or may be partially defined, and that the choice of groups can have a large impact on the resulting explanation. We hope our framework can be a building block for future research on methods for automatically selecting groups or selecting groups with much less supervision.
> >
> > ## Relation between spurious correlations and shift explanations
> > Spurious correlations are often defined as a relationship between variables that is not causal, which is a general definition depending on the choice of variables, so your question raises the interesting point of what a spurious correlation is for explaining distribution shift. One definition is to consider spurious correlations in regards to the source vs. target membership itself. For instance, we may determine that race is spuriously correlated with the target distribution and the shift is actually due to shifts in education level and marital status. In this case, a regular shift explanation may explain the shift using the spurious race correlation, leading to group irregularities since the explanation will fail for the subpopulation without the spurious correlation present. In addition, if the spurious correlation is with an unactionable feature, then the shift explanation will be infeasible since it will use the unactionable spurious feature to explain the shift. We can use GSE to learn an explanation without this spurious correlation by choosing groups so that one group does not have the spurious correlation present.
> >
> > Therefore, the definition of spurious correlations can be reduced to infeasible explanations by defining infeasible features as the features without a causal role in the distribution shift. In this case, an infeasible explanation must exploit one of the non-causal features to explain the distribution shift, meaning that it has learned a spurious correlation. By using our method with groups defined by the spurious features, we can reduce the amount of spurious features in the learned explanation.

---

> > > ### Author Response · Authors · 2023-11-22
> > >
> > > We believe we have addressed all your concerns, but if there is anything remaining, we are happy to respond in the remaining discussion period. We would also appreciate any consideration of adjusting your score in light of the improvements and clarifications made in our response. Thank you!

---

> > > > ### Comment · Reviewer_XuMW · 2023-11-23
> > > > **Official Comment by Reviewer XuMW**
> > > >
> > > > Thank you the authors for the response and efforts. I would like to keep my score as is.

---

### Author Response · Authors · 2023-11-18

Thank you for reading our paper and giving helpful feedback! We are especially happy to see that reviewers AAG5 and LTxx found the problem interesting and important and reviewers XuMW and LTxx found the empirical evaluation extensive. We address comments and questions that were shared among more than one reviewer below:

## Novelty of our method
In response to the comments about the novelty of our method from Reviewers XuMW and LTxx, we want to emphasize that our goal is very different from improving model robustness or developing and applying DRO methods. Our goal is to reveal that distribution shift explanations (which explain shifts in data independent of model performance) can be poor for subgroups and we propose a worst-group loss as a way to mitigate the issue. In addition, we perform a theoretical analysis and provide empirical evidence that a worst-group loss can improve shift explanation feasibility and robustness, which has not been done before to the best of our knowledge.

## Updates to the theoretical analysis
Based on the feedback from Reviewers XuMW and 3qo2, we have revamped Theorem 1 in the paper to not use preselected numbers. We also included two more theorems which analyze feasibility and robustness of explanations when they are learned by optimizing reduction in Wasserstein distance compared to reduction in Wasserstein distance for the worst-group.

## Requirement of group annotations
Reviewers XuMW and AAG5 commented that our method requires group annotations, but we want to clarify that our method does not depend on human annotated group labels. We chose to focus on the case when group labels are available, since it is often the case that group labels are present [Sagawa et al, ICLR 2020; Zhang et al, ICLR 2021; Kirichenko et al, ICLR 2023]. In Appendix E (mentioned in Section 4.1) we discuss when group labels are not present and perform an experiment without group labels where GSE still improves feasibility and robustness. To get group labels without supervision we use a clustering technique to define groups based on feature space similarity.

[Sagawa et al, ICLR 2020]: Sagawa, S., Koh, P. W., Hashimoto, T. B., and Liang, P. Distributionally robust neural networks for group shifts: On the importance of regularization for worst-case generalization. In International Conference on Learning Representations (ICLR), 2020.

[Zhang et al, ICLR 2021]: Zhang, Jingzhao, et al. "Coping with Label Shift via Distributionally Robust Optimisation." International Conference on Learning Representations. 2021.

[Kirichenko et al, ICLR 2023]: Kirichenko, Polina, Pavel Izmailov, and Andrew Gordon Wilson. "Last Layer Re-Training is Sufficient for Robustness to Spurious Correlations." The Eleventh International Conference on Learning Representations. 2023.

---

### Meta-Review · Area_Chair_ZZNJ · 2023-12-04

**Metareview:**

This work proposes Group-aware Shift Explanations (GSE), a shift explanation method for understanding distribution shifts. For distributions covering changeable and unchangeable properties, computing the shift blindly will lead to infeasible solutions. This is avoided by grouping the samples, i.e, maintaining the unchangeable properties as constraints. The proposed method is evaluated on several datasets, most of which containing language data or based on language data.

Strengths
* first shift explanation method
* extensive experiments

Weaknesses
* strong assumptions without references
* novelty "the submission is different from the worst-group robustness methods, but the main contribution is still just incorporating the worst-group loss for explaining distribution shifts."
* presentation of the paper "the revised manuscript is slightly easier to understand, however, it is still quite difficult for readers to understand the goal of the proposed method intuitively"

Initially Reviewer 3qo2 raised some ethical concerns about "sex being considered to be a permanent feature." The rebuttal/revision addressed this particular concern.

**Justification For Why Not Higher Score:**

Three expert reviewers recommended rejection and the positive reviewer was also concerned about the novelty and presentation of the submission.

**Justification For Why Not Lower Score:**

NA

---

### Decision · Program_Chairs · 2024-01-16

Reject